

# Seasonal to decadal dynamics of supraglacial lakes on debris-covered glaciers in the Khumbu Region, Nepal

Lucas Zeller[1], Daniel McGrath[1], Scott W. McCoy[2], and Jonathan Jacquet[2]

[1]Department of Geosciences, Colorado State University, Fort Collins, CO 80523, USA
[2]Department of Geological Sciences and Engineering, University of Nevada, Reno, NV 89557, USA

**Correspondence:** Lucas Zeller (Lucas.Zeller@colostate.edu)

**Abstract.** Supraglacial lakes (SGLs) play an important role in debris-covered glacier (DCG) systems by enabling efficient interactions between the subglacial, englacial, and supraglacial environments. Developing a better understanding of the short-term and long-term development of these features is needed to constrain DCG evolution and the hazards posed to downstream communities, ecosystems, and infrastructure from rapid drainage. In this study we present an analysis of supraglacial lakes

on eight DCGs in the Khumbu region of Nepal by automating SGL identification in PlanetScope, Sentinel-2, and Landsat 5-9 images. We identify a regular annual cycle in SGL area, with lakes covering approximately twice as much area during their maximum annual extent (in the pre-monsoon) than their minimum (in the post-monsoon season). The high spatiotemporal resolution of PlanetScope imagery (∼daily, 3 meter) shows that this cycle is driven by the appearance and expansion of small lakes in the upper debris-covered regions of these glaciers throughout the winter. Decadal-scale expansion of large, near-

terminus lakes was identified on four of the glaciers (Khumbu, Lhotse, Nuptse, and Ambulapcha), while the remaining four showed no significant increases over the study period. The annual variation in SGL area is of comparable or greater magnitude as decadal-scale changes, highlighting the importance of accounting for this seasonality when interpreting long-term records of SGL changes from sparse observations. The complex spatiotemporal patterns revealed in our analysis are not captured in existing regional-scale glacial lake databases, suggesting that more targeted efforts are needed to capture the true variability of

SGLs on large scales.

## 1 Introduction

The processes controlling the response of debris-covered glaciers (DCGs) to changing climatic conditions are distinct from those of clean-ice glaciers and tend to be less well studied and understood (Brun et al., 2019; Miles et al., 2020). Supraglacial lakes (SGLs) on DCGs can increase ablation rates through increased radiation absorption, ice cliff calving, and efficient transfer

of heat to the englacial and subglacial environments (Miles et al., 2022). Further, rapid drainage of these lakes present a potential hazard to downstream communities and infrastructure (Miles et al., 2018; Rounce et al., 2017), particularly in cases where smaller ponds coalesce into large terminal lakes capable of causing catastrophic glacier lake outburst floods (Watanabe et al., 2009).





Understanding the evolution of SGLs on DCGs has been limited by difficulties associated with directly observing these
features (Racoviteanu et al., 2022). The rugged topography of DCGs generally limits the scope of in situ investigations to
single lakes and/or individual glaciers. Remote-sensing approaches to studying the glacier-wide spatial and temporal patterns
in lake evolution have been limited by 1) the spatial resolution of freely available satellite imagery (e.g., Landsat and Sentinel-
2) being large relative to the size of many SGLs, 2) the infrequent revisit period of satellites acquiring higher spatial resolution
imagery preventing detailed assessment of the rapid temporal evolution of these lakes.

Previous studies have found that SGLs tend to form in areas with low surface gradients and low glacier flow velocities
(Reynolds, 2000). The temporal stability of individual lakes can vary widely, with some appearing stable on decadal scales,
while others fill and drain annually, semi-annually, or seemingly randomly (Narama et al., 2017; Steiner et al., 2019). Seasonal-
to-annual patterns in lake development have been observed, with links to ice velocity variations, water availability from precip-
itation and snow/ice melt, and the opening or collapsing of englacial conduits (Benn et al., 2001; Mertes et al., 2017; Watson
et al., 2018b).

There are few existing studies which have investigated the seasonal variations in SGL lake area on DCGs on a glacier-wide
scale. These studies have generally relied on coarser-resolution Landsat imagery aggregated over decadal time periods (Miles
et al., 2017; Narama et al., 2017), or have used a sensor integration approach with both optical and synthetic aperture radar
(SAR) datasets (Wendleder et al., 2021). Further, the majority of these studies (including ours) have focused on DCGs in
High Mountain Asia, and particularly in regions of the Himalaya and Karakoram which are impacted by the Indian summer
monsoon.

These studies find that SGLs generally form in the early spring from the accumulation of rain and meltwater, grow in size
through the pre-monsoon season (1 March–15 June), and then drain during the monsoon season (15 June–30 September). The
frequent snow cover and frozen surfaces tend to make lake identification difficult during the winter, but SGLs are generally
assumed to be stable or very slowly drain throughout the post-monsoon (1 October–30 November) and winter (1 December–28
February) months (Watson et al., 2018b; Miles et al., 2017).

Despite these patterns in SGL area and drainage that have been observed over seasonal timescales, the majority of studies
investigating SGL development on decadal timescales have not explicitly accounted for seasonality in lake area (e.g., Mohanty
and Maiti, 2021; Watson et al., 2016; Steiner et al., 2019). Rather, these studies tend to utilize a series of individual higher-
resolution images that cover a wide range of years but are captured at irregular seasonal timing.

The goal of this study is to develop consistent observations of SGL evolution over daily-to-decadal timescales. To do so, we
integrate PlanetScope satellite imagery (Planet Team 2017), Sentinel-2 and Landsat 5–9 imagery, and in-situ field observations
to investigate SGL dynamics on eight glaciers in the Khumbu region of Nepal.

## 2   Study Area

We investigate SGL dynamics on eight glaciers in the Khumbu region of Nepal (Figure 1; Table S1). The glaciers in this
region are characterized by ablation zones which have nearly continuous debris cover, slow ice flow velocities (<10 m/yr), and



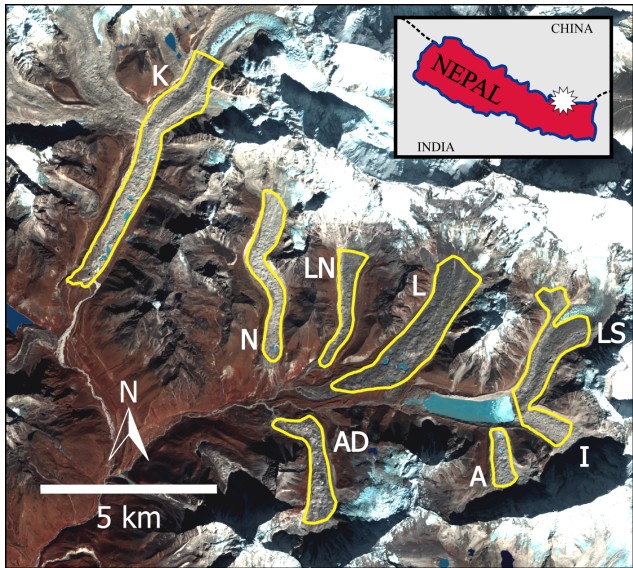

**Figure 1.** Sentinel-2 satellite image (13 November 2020) of the study area. The debris-covered areas of the eight glaciers are outlined in yellow. Letters adjacent to each glacier are shortened names (K=Khumbu, N=Nuptse, LN=Lhotse Nup, L=Lhotse, LS=Lhotse Shar, I=Imja, A=Ambulapcha, AD=Ama Dablam). Inset map shows the region's location within Nepal (indicated by the white star).

low-gradient surface slopes (<5°). Debris-covered areas range in elevation from 4800–5300 m, with debris-thickness tending to increase towards the glacier termini (Rounce et al., 2021). Five of the glaciers in this study have a southerly aspect and flow direction, two (Ambulapcha and Ama Dablam) flow north, and one (Imja) flows predominantly westward (Figure 1). Lhotse
Shar and Imja flow into the rapidly-expanding Imja Tsho, resulting in increased ice flow velocities compared to the other glaciers in this study (Watanabe et al., 2009).

The climate in the Khumbu region is characterized by monsoonal summer seasons, during which the majority of annual precipitation falls, and cold and dry winter seasons (Perry et al., 2020). The limited winter precipitation leads to non-continuous snow cover on the study glaciers' surfaces throughout the winter, however cold air temperatures regularly resulted in frozen
lake surfaces, particularly in the February–April time period.

## 3   Methods

In this study, we investigated the patterns of SGL development on time scales ranging from subseasonal to decadal. We achieved this by integrating multiple satellite remote sensing imagery sources. Each individual source allows unique observations, while together they build upon each other to inform us on their relative strengths and limitations. Our methodology is broadly broken
into a two-step process built around identifying SGLs in PlanetScope, Landsat, and Sentinel-2 imagery. We first developed a novel approach to delineating SGLs in high temporal resolution time series of PlanetScope imagery. From this dataset, we investigated the seasonal patterns in SGL variations over the 2017–2022 time period. We then used these high spatial



resolution observations (3 m scale) as a validation dataset to optimize SGL identification in coarser-resolution Landsat and Sentinel-2 imagery, allowing us to investigate the decadal-scale trends in SGL evolution at coarser spatial scale (10 m and 30 m). Direct field observations and Unmanned Aerial System (UAS) surveys from May 2023 provided additional insight into the variable spatial and temporal patterns of SGL development which are found in the study area.

The use of high resolution PlanetScope imagery allowed us to identify supraglacial streams and very small lakes which may be more appropriately referred to as "ponds". There is no clear consensus in previous literature on the differentiation between terming a supraglacial feature a "lake" versus a "pond", and without in situ observations it is difficult to differentiate between a supraglacial "stream" which is flowing and stagnant ponded water. Therefore, in this paper we use the term "supraglacial lake" (SGL) to refer to any supraglacial hydrologic feature which we identify from satellite imagery.

The area of investigation (AOI) was limited to the debris-covered tongue of each glacier. These outlines were created by manually adjusting the glacier extents from the Randolph Glacier Inventory (RGI Consortium 2017) to better match glacier margins in high-resolution satellite imagery in Google Earth Pro (sourced from Maxar imagery). Areas of exposed glacier ice were excluded as they were frequently mis-classified as water. For this analysis, we separated Lhotse Shar and Imja into two glaciers, although they are classified as a single glacier in the Randolph Glacier Inventory. The outlines used for these two glaciers excluded the rapidly expanding terminal lake Imja Tsho for the entire time period, as the focus of this study was solely on SGLs.

## 3.1 SGL identification: Planet

We developed a largely-automated workflow to identify lakes on DCGs in PlanetScope images (Figure 2). We applied this workflow over eight DCGs in the Khumbu region of Nepal over the October 2017 to April 2022 time period (Table S1, Figure 1). We identify lakes with ice or snow-covered surfaces in addition to unfrozen lakes by taking advantage of the high temporal resolution of the imagery (see Section 3.1.2).

### 3.1.1 Data access and cleanup

Imagery covering the AOI was searched for and downloaded using the Planet Orders API (Planet Team 2017). For each glacier, we selected images that (1) captured the entire glacier extent, (2) had less than 15% cloud cover across the entire image, and (3) contained four-band (RGB and near-infrared (NIR)), ortho-rectified, surface-reflectance data (the "ortho_analytic_4b_sr" asset type). We additionally included pairs of images which were taken by the same satellite on the same day that, when combined, captured the entire glacier surface and each met criteria (2) and (3). A full list of all images meeting these criteria was compiled for the eight study glaciers. The 'harmonize' tool was used when ordering these images using the Planet Orders API. This tool brings the spectral response of each image in line with coincident Sentinel-2 imagery, ensuring a consistent spectral response across the multitude of individual sensors.

This entire image collection was downloaded to a local computer for additional processing and lake identification. For each glacier, images were merged together (if needed) and then clipped to the AOI extent. Each image was then manually inspected



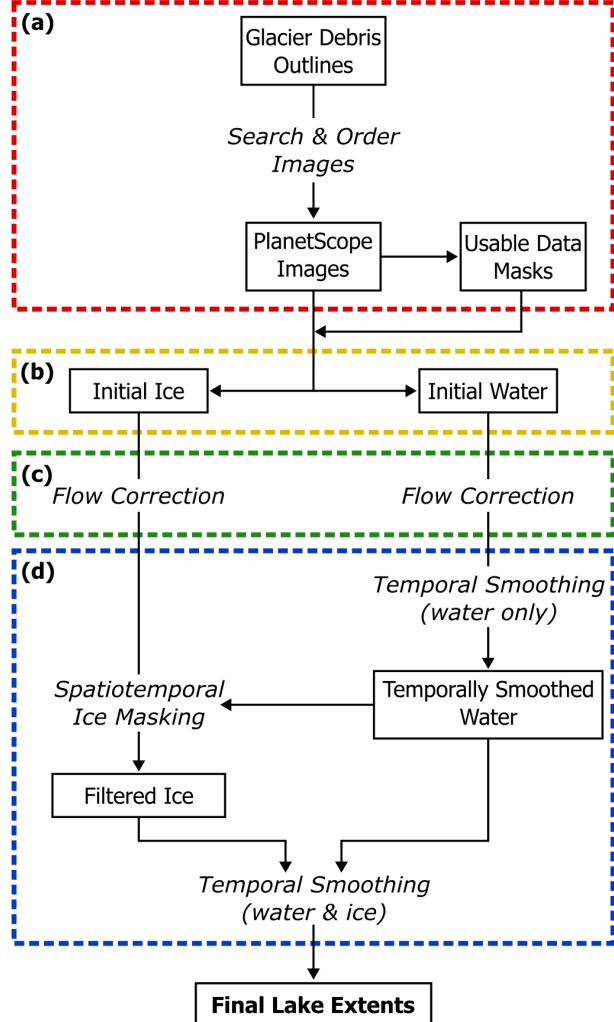

**Figure 2.** Workflow diagram for delineating supraglacial lakes in PlanetScope imagery. The boxed items indicate intermediate products in the workflow, while unboxed, italicized labels indicate processing steps which are refered to in the text.

to ensure suitable data quality. Images with considerable cloud cover or snow cover, anomalous spectral responses, or poor band coregistration were discarded. This was the only manual intervention in the otherwise fully automated workflow.

    Areas of cloud cover and terrain shadows were identified in each image and excluded from future analysis. Cloud-covered areas were classified using the Planet-provided usable data mask. We developed an approach to identify terrain shadows within PlanetScope images as dark, spatially-continuous areas. Further details on the shadow masking can be found in the
supplementary material.



### 3.1.2 Initial lake identification

SGLs were identified by separately classifying water and frozen lakes, and then applying a pixel-wise temporal smoothing algorithm to ensure a consistent evolution of the glacier surface. An initial classification of water (Figure 2b) was performed by using a spatially-varying threshold on the normalized-difference water ratio (NDWI, green minus near-infrared (NIR) di-
vided by green plus NIR; (Gao, 1996)), which was calculated after applying scaling factors to convert band values to surface reflectance. Each pixel was then classified as water if its NDWI was at least 0.10 greater than the mean NDWI of surrounding pixels within 150 meters (not including off-glacier areas, cloud cover, and terrain shadows) and with a minimum NDWI threshold of –0.15. A final shadow masking step was also applied at this point, with pixels with average surface reflectance across visible (RGB) bands less than 0.02 excluded from being classified as water.

An initial classification of frozen lake surfaces (Figure 2b) was performed using a similar spatially-varying threshold on the NIR band. Each pixel was classified as frozen water (ice) if the NIR digital number was at least 1000 greater than the mean NIR of surrounding pixels within 150 meters (not including off-glacier areas, cloud cover, and terrain shadows). In simple terms: areas of relatively high NDWI were initially classified as water, while areas of relatively high brightness in the NIR band were initially classified as lake ice. This initial ice classification does not differentiate between lake ice, glacier ice, and snow,
however we take precautions to avoid including glacier ice and snow in the final products (described in Section 3.1.3). The study extent was delineated to purposefully exclude areas of the glaciers that were not completely debris-covered, reducing the possibility of glacier ice being mis-classified as lake ice. Images with widespread snow cover were excluded (Section 3.1.1). Additionally, in later steps we only allow areas which were recently classified as water to be classified as frozen lakes (Section 3.1.3), limiting the effects of mistakenly identifying snow drifts or exposed ice cliffs as SGLs.

An ice flow correction was applied to each initial water and lake ice classification image to align them to a common date of 1 January 2020 (Figure 2c). This removes the effects of lake advection due to ice flow and allows multi-year tracking of individual lakes. Glacier flow velocities were taken from 1985–2022 composite ITS_LIVE velocity mosaics (Gardner et al., 2023) resampled to PlanetScope resolution.

### 3.1.3 Filtering and Smoothing

Following the initial classification of SGLs there was significant variability and noise in the time series due to variable image quality, occasional snow and cloud cover, and inconsistent spectral response across images. In order to reduce this noise and to ensure a consistent temporal evolution of SGLs in our products, we applied a series of filtering and smoothing steps (Figure 2d).

A pixel-wise temporal smoothing algorithm was applied to the initial water classification images to prevent individual
pixels from rapidly switching between being classified as water and non-water. For each pixel, we collated the time series of all observations across the 2017–2022 period. Individual observations were assigned a value of 1 if the pixel was identified as water and 0 if it was not. A Gaussian filter was then applied to this time series, using a Gaussian kernel with a standard deviation of 14 days. For each point, if the resulting smoothed value was greater than or equal to 50% of the potential maximum value





(the value it would be if all observations were initially classified as water) then that pixel in that image was given a "smoothed"
classification of water. All other observations were classified as non-water. The products at the end of this smoothing process
(Figure 2d, Temporally Smoothed Water) represent the spatiotemporal distribution of unfrozen SGLs.

The temporally smoothed water products were then used to further constrain the classification of frozen lake surfaces (Figure
2d, Spatiotemporal Ice Masking). Each pixel which was initially classified as lake ice remained as a lake ice pixel only if it
was also classified as water at any point within the preceding or following 60 days (resulting in the Filtered Ice product). This
limited the mis-classification of snow drifts and ice cliffs as SGLs.

Finally, the filtered ice dataset was combined with the temporally smoothed water dataset, and the same Gaussian temporal
smoothing was applied to this combined dataset, resulting in the final lake extent dataset. This provides the extent of all lakes,
including frozen and unfrozen, within each image.

### 3.2 SGL identification: Landsat and Sentinel-2

#### 3.2.1 Data access and cleanup

The Planet-derived SGL products were used as a training dataset for developing an optimized SGL identification approach
for Landsat and Sentinel-2 imagery in our study area. All Landsat 5, 7, 8, 9 and Sentinel-2 imagery, spanning the years
1988–2022, was downloaded from Google Earth Engine (n=1230 per glacier). We used Landsat Tier 1 (Collection 2) surface
reflectance products and Level-2A harmonized surface reflectance Sentinel-2 images. The same debris-covered AOIs were
used for all images to facilitate a direct comparison between all satellite sources and across the entire 35-year time frame.
Each image of each glacier was manually checked, and images with significant cloud cover, snow cover, or shadows over the
majority of the glacier were discarded, resulting in an average of 940 images remaining for each glacier (Table S2). Shadowed
areas in each image were masked using the Google Earth Engine ee.Terrain.hillShadow() function applied to the NASADEM
(NASA JPL 2020) with image-specific sun azimuth and elevation as input. Additional remaining shadows were removed by
thresholding on surface reflectance values in the green band (using values of 0.08 for Landsat images and 0.11 for Sentinel-2).
Persistent exposed glacier ice, which was frequently present in older Landsat imagery and mis-identified as water, was masked
as described in the supplementary material.

#### 3.2.2 Validation dataset creation

All Landsat 7, Landsat 8, and Sentinel-2 images with coincident Planet-derived lake extents were identified to be used as a
training dataset. We defined suitable images as those which had at least three Planet-derived observations within the nine-day
period centered on the Landsat/Sentinel-2 image date. For each of these training images, we resampled the corresponding
Planet-derived lake extents to give a 30 m (Landsat) or 10 m (Sentinel-2) resolution product that indicated the percent water
cover within each Sentinel-2/Landsat pixel (Figure 3). For lake identification in Landsat and Sentinel-2 imagery, we focused on
only unfrozen lakes, rather than also attempting to identify frozen lake surfaces as we did in PlanetScope imagery. The reason
for this was that the limited temporal resolution of Landsat and Sentinel-2 did not allow for the same frozen water identification



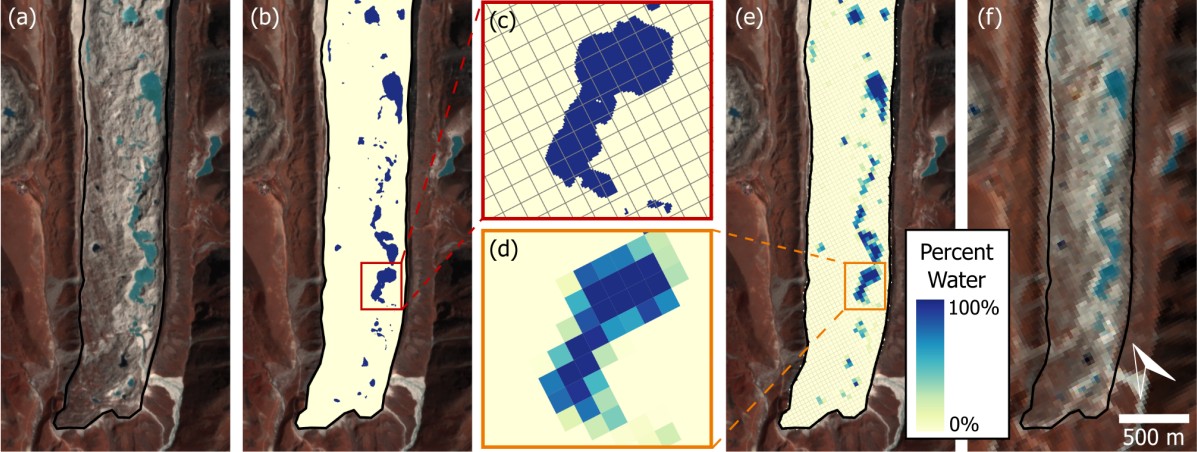

**Figure 3.** An example of the process of creating a Landsat-scale training dataset from Planet-derived lake extents. (a) shows an example PlanetScope image of the terminus of Khumbu Glacier on 24 October 2018, with the glacier extent outlined in black. (b) shows the lake extents in blue, from the automated PlanetScope workflow. (c) shows the Landsat pixel grid overlaid on the lake extents, for a subset of the glacier area. (d) shows the same area, resampled to Landsat pixel size and grid, colored by the percent of water cover within each pixel. (e) shows the full terminus area, resampled to the Landsat grid, colored by percent water coverage. (f) is a Landsat image from the same date (24 October 2018)

approach to be employed. Thus, the Planet-derived products we used for developing the training dataset include only the lakes identified without employing the frozen lake identification (i.e., we used the temporally smoothed water product in Figure 2d).

### 3.2.3 NDWI optimization and lake identification

Using the Planet-derived training dataset, we identified the optimal NDWI threshold (calculated after scaling images to surface
180 reflectance) to use for binary classification of SGLs in Landsat and Sentinel-2 imagery (Watson et al., 2018a). For each sensor (Landsat 7, Landsat 8, and Sentinel-2) a single NDWI threshold was identified that minimized the difference between Planet-derived and Landsat/Sentinel-derived total lake area across the entire training dataset. There were no Landsat 5 images in the training dataset because the Landsat 5 mission ended in 2012. Landsat 9 images were included in our analysis, but there were insufficient images overlapping the PlanetScope validation dataset to reliably constrain an optimized NDWI threshold. Thus,
185 the same NDWI threshold for Landsat 7 was used for Landsat 5 images, and the same threshold for Landsat 8 and 9, because the sensor band wavelength ranges are identical.

These optimized NDWI thresholds (Table 1) were then used to identify SGLs on the glacier surfaces across all available imagery, after removing shadowed areas and exposed glacier ice (Section 3.2.1). No additional smoothing was applied to these products, due to the limited temporal resolution compared to PlanetScope imagery.



**Table 1.** The optimized NDWI threshold that was found for each Landsat and Sentinel-2 sensor.

| Satellite | Optimized NDWI |
|---|---|
| Landsat 5 & 7 | 0.172 |
| Landsat 8 & 9 | 0.226 |
| Sentinel-2 | 0.260 |

## 3.3 Field observations

A field campaign was undertaken in May 2023 to collect in-situ and drone-based observations of the study area. Perspective drone flights were flown on five glaciers (Lhotse Shar, Imja, Ambulapcha, Lhotse, and Lhotse Nup) to capture high resolution videos and photos of the debris-covered surfaces using a DJI Mavic 2 Pro and DJI Mavic Air 2s. Additionally, grid surveys were flown near the termini of Ambulapcha, Lhotse Nup, and Khumbu glaciers to create high-resolution Structure from Motion Multi-View Stereo (SfM) digital elevation models (DEMs) and orthomosaics. Grid surveys were flown ~60–90 meters above ground level, with nadir camera positioning and at least 70% overlap between images. SfM processing was completed in Agisoft Metashape. Ground control points (GCPs) were placed and surveyed for all three grid surveys, however there were no nearby operational base stations to use for PPK processing of these points, so they were not used in model creation. The model creation resulted in DEMs with 0.08 m spatial scale and orthomosaics with 0.04 m spatial scale.

## 4 Validation

A validation set for the Planet-derived lake extents was created by manually delineating SGLs in a total of 28 PlanetScope images of Lhotse, Nuptse, and Ambulapcha glaciers (Figure S1). For each glacier, a series of cloud-free images spanning a range of approximately one year were selected, in order to capture the full annual cycle of SGL expansion and drainage. We used a ±1 pixel (3 m) buffer for error estimation of these manual delineations (Gardelle et al., 2011). Further, the total SGL area on seven glaciers which were manually mapped by Watson et al. (2016) was compared to our results from concurrent months. While there is no overlap in the years investigated between our study and theirs (the latest imagery used in Watson et al. (2016) was from 2015), it provides a valuable constraint on the expected magnitude of SGL area.





# 5 Results

## 5.1 Lake delineation accuracy

### 5.1.1 Planet

The Planet-derived SGL extents resulted in physically realistic, accurate delineations of SGLs at high spatiotemporal resolution (Figure 4, Figure 5). The mean absolute error in total SGL area was 0.039 km$^2$ for Lhotse Glacier, 0.018 km$^2$ for Nuptse Glacier, and 0.004 km$^2$ for Ambulapcha Glacier. These are equal to errors of 0.75%, 0.66% and 0.41% of the total debris-covered area (from our AOIs) of each glacier respectively. During the months with highest temporal density of suitable imagery (September–February), these errors are significantly lower (0.19%, 0.36%, and 0.28% of debris-covered area, respectively). Further, our automated SGL areas showed good agreement with results from Watson et al. (2016) at similar annual timing (Figure S1), with the majority of their results falling within our inter-annual range of values. A notable difference can be observed in the lake area on Khumbu Glacier, where our results show a greater SGL area than observed by Watson et al. (2016) which is likely due to the well-documented long-term expansion and coalescence of lakes near the terminus (Naito et al., 2000). Similarly, the large differences on Imja Glacier may be attributed to changes in subglacial hydrology as the proglacial lake Imja Tsho expands (Bolch et al., 2008).

### 5.1.2 Landsat and Sentinel-2

The optimized NDWI thresholds for lake identification were found to be 0.172 for Landsat 5 and 7, 0.226 for Landsat 8 and 9, and 0.260 for Sentinel-2 images (Figure 6, Table 1), with total errors across the entire validation dataset being less than 1%. The optimized value for Sentinel-2 is similar to the value found by Watson et al. (2018a) of 0.22. The optimized value they found for Landsat 8 is not directly comparable to our results because they used the modified NDWI (Xu, 2006) which uses a short wave infrared band in place of the near infrared band when calculating NDWI.

The average per-image glacier-wide error in SGL area, compared to the coincident Planet-derived lake area, was 0.49% for Landsat and 0.43% for Sentinel-2 (presented as a percentage of total debris-covered area), with values for individual glaciers ranging from 0.12%–0.99%, and 0.07%–1.04% respectively (Table S3).

## 5.2 Short term dynamics

### 5.2.1 Seasonality in glacier-wide lake area

A pronounced seasonal cycle in total SGL area was found in the Planet-derived lake datasets (Figure 7, Figure S1). On most glaciers, the maximum glacier-wide SGL area (both frozen and open water) occurred in March (during the pre-monsoon season), decreased throughout the monsoon with a minimum area occurring in September, and then increased steadily throughout the post-monsoon and winter months (October–February) (Figure 7). For each glacier, the annual maximum SGL area is approximately twice the annual minimum, highlighting the magnitude of short-term variability these features can exhibit.



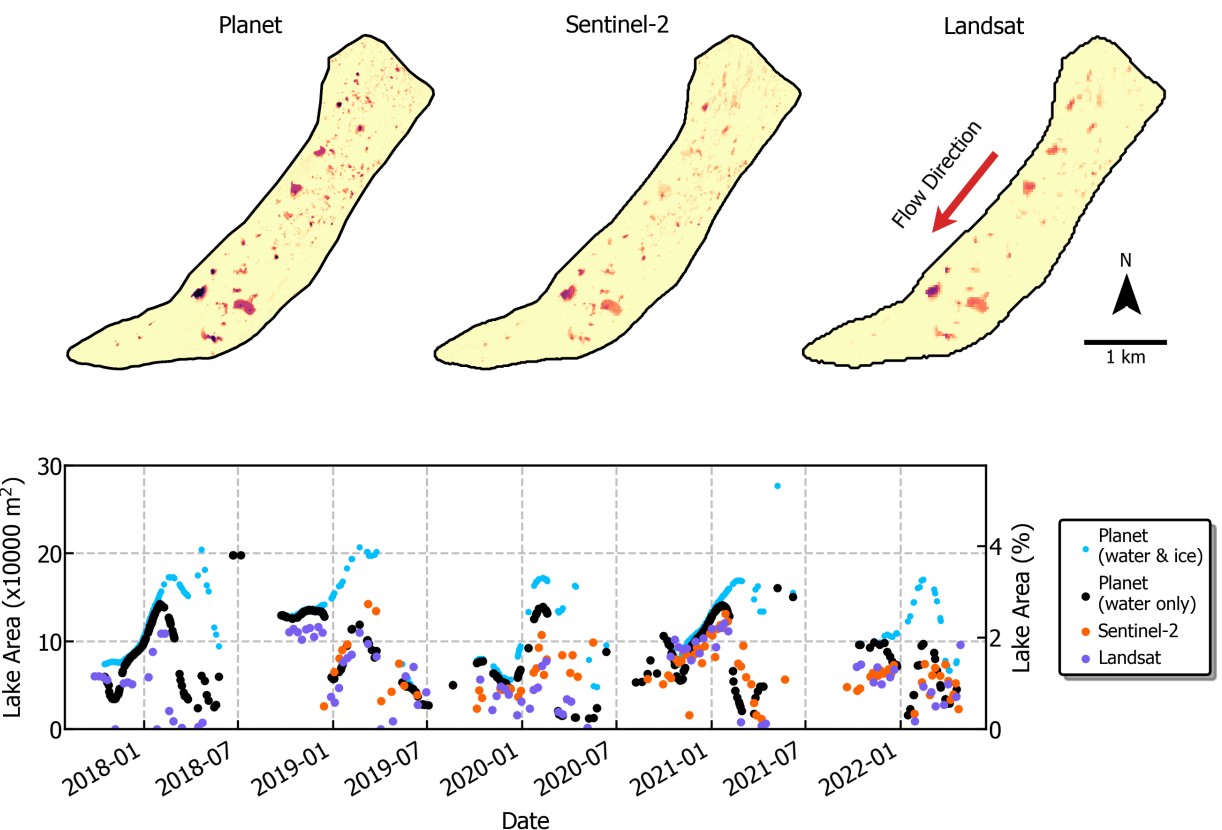

**Figure 4.** A comparison between the SGL extents and frequency in Planet, Sentinel-2 and Landsat imagery on Lhotse Glacier. The colorscale corresponds to the frequency each pixel was classified as water, with darker colors being more frequent. Only Landsat images from the 2018–2022 time period were used, to provide a more direct comparison with Planet and Sentinel-2. The time series plot shows the trend in total lake area for each sensor. Each point represents water identified in a single image. Planet-derived lake areas are shown for both the full methodology which includes frozen lake area (cyan) and the area if frozen lake area is excluded (black).

While this cycle is most clearly seen in the high temporal resolution Planet-derived dataset, it is also apparent in Sentinel-2 and Landsat-derived time series, although frequent lake surface freezing in winter months (which was not classified in Landsat/Sentinel-2) obscures this trend in many years (Figure 4).

The winter-time increase in SGL area is not caused by an increase in the frozen SGL area (Figure S2). We find that the maximum frozen lake surface extent occurs in April, and minimal lake surface freezing occurs in the September–December time period during which unfrozen lake area is steadily increasing (Figure S2).

A detailed investigation of the five-year Planet-derived SGL inventory shows that the seasonal cycle in total SGL area (both frozen and unfrozen) is driven by the regular appearance and expansion of small lakes and supraglacial streams throughout the



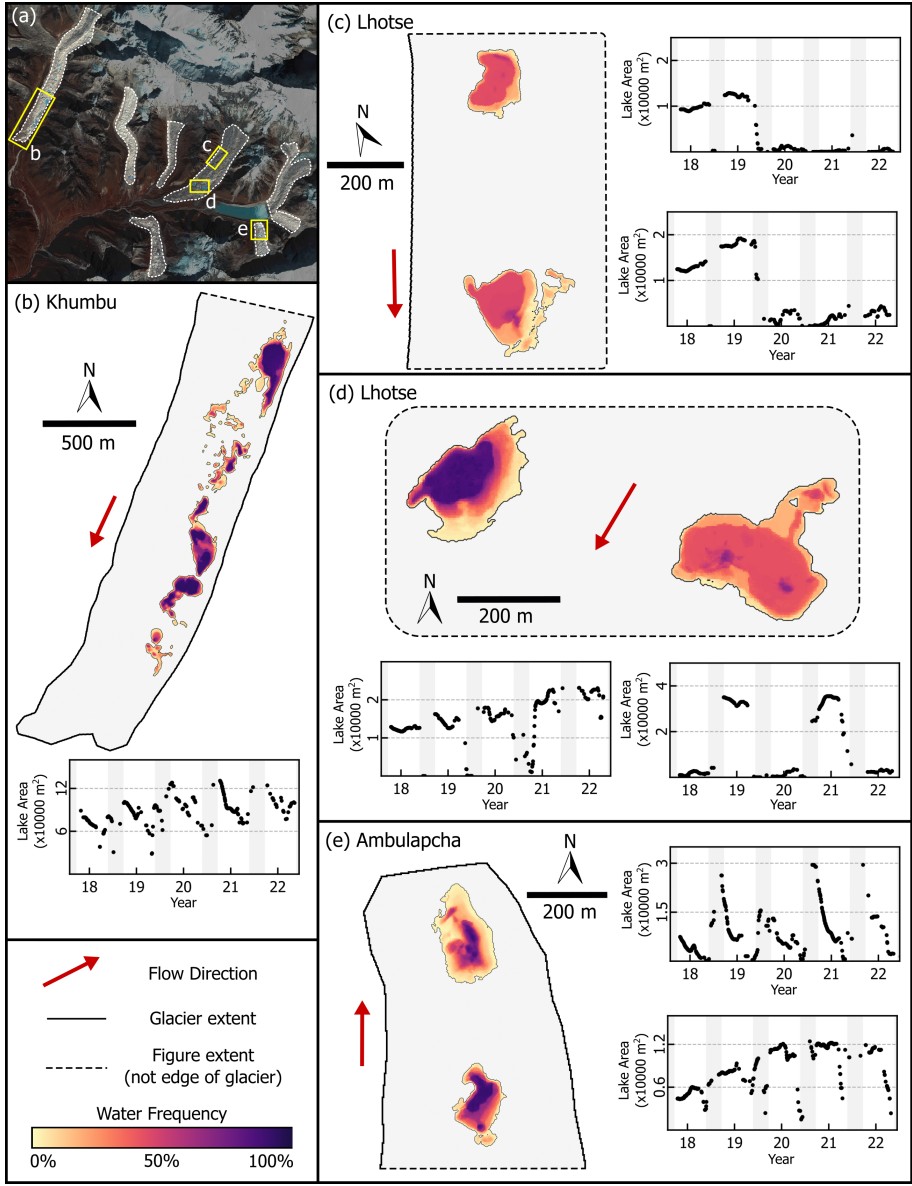

**Figure 5.** Water frequency maps and time series of select individual lakes or groups of lakes. Locations of each (b-e) are indicated in the top-left overview map. Lake maps are colored by water frequency (darker colors indicating more frequent water). The time series show the total area in individual Planet images. Grey shaded areas indicate monsoon season (15 June-30 September). Year tick-marks indicate the start of each year. Red arrows indicate the glacier flow direction.

post-monsoon and winter months in the upper debris-covered regions further from the glacier termini (Figure 8), rather than seasonal growth/shrinking of the larger lakes in the study region.





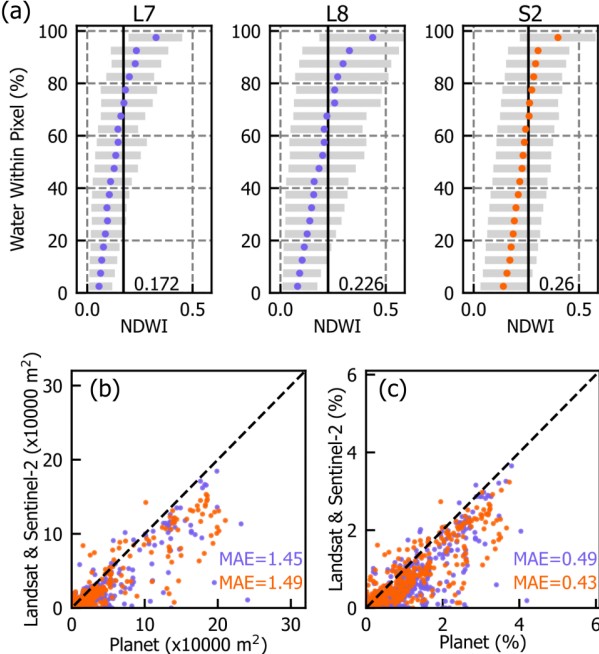

**Figure 6.** (a) NDWI optimization results for each satellite. Each horizontal bar shows the range of NDWI values for Landsat and Sentinel-2 pixels in the validation dataset with the indicated percent water cover (aggregated in 5% bins). Dots indicate the mean, grey bars show the range containing 90% of all observations. Horizontal lines and the shown number indicate the optimized NDWI value for each sensor, which minimized the net error over the validation dataset. (b) and (c) compare the SGL area identified in PlanetScope imagery and in Landsat/Sentienel-2 coincident imagery using the optimized thresholds, with each point represetning a single image. (b) shows these values as the total lake area, while (c) scales them by the total glacier debris-covered area. Mean absolute error (MAE) is shown for Landsat (purple) and Sentinel-2 (orange) results on each plot.

Between October and February in each of the five years on Lhotse Glacier, lake area increase was greater in regions further from than terminus (Figure 8a). Similarly, the total number of lakes increases throughout these months each year, with this

driven predominantly by the increasing number of small lakes (lakes between 45 and 450 m$^2$, or less than half of a single Landsat pixel) (Figure 8c). Lastly, the relative contribution of smaller lakes to the total lake area increases between October and February each year (Figure 8b). For example, lakes which are smaller in size than a single Landsat pixel (900 m$^2$) account for an average of 18% of total lake area on Lhotse Glacier in October, but account for 34% of total lake area on average in February (Figure 8c).

Taken together, these observations show that the increase in lake area between October and February is predominantly due to the appearance and expansion of small lakes and/or streams, rather than changes in area of larger lakes. This same pattern is observed (to varying degrees) on most glaciers in our study area, particularly on the four glaciers with the greatest lake area (Khumbu, Nuptse, Lhotse, and Lhotse Shar) (Figure 7, Figures S3–S10).



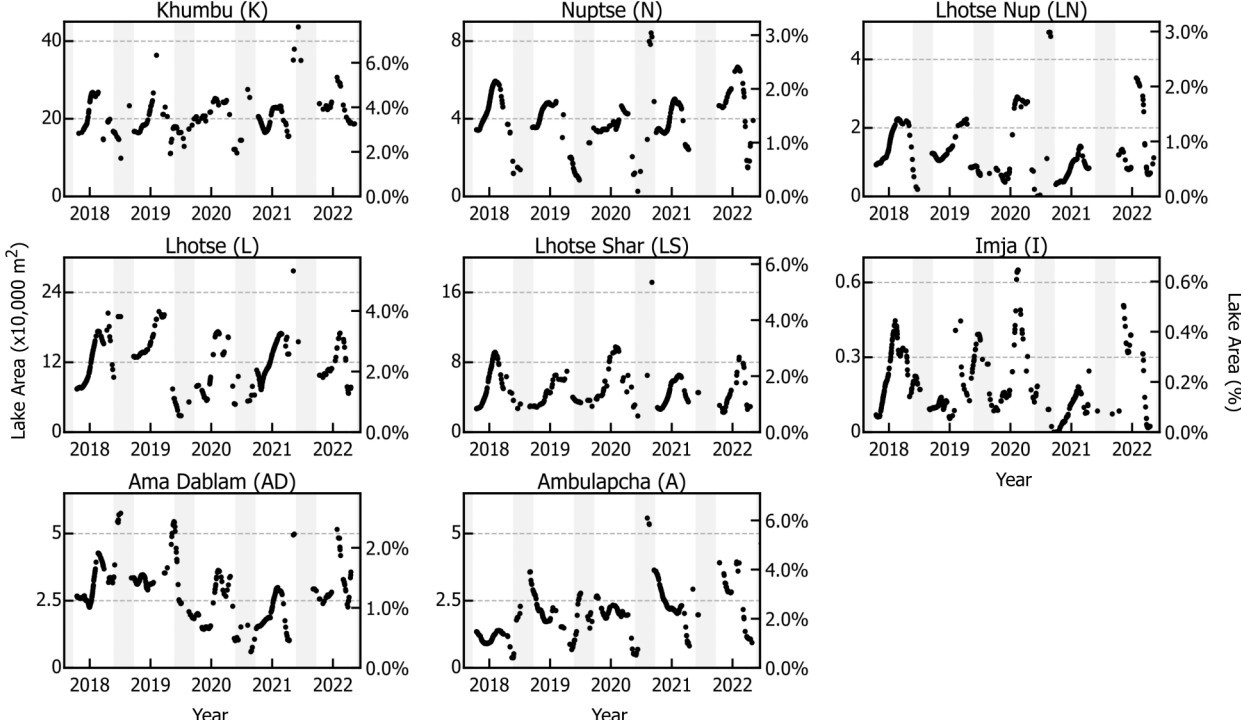

**Figure 7.** Time series of total SGL area on each glacier from PlanetScope imagery. Each point indicates the total lake area identified from the automated methodology. Grey shaded areas indicate monsoon season. Left axis of each plot shows lake area, while right axis shows the values scaled to a percent of the total debris-covered area.

### 5.2.2 Individual lake dynamics

The high spatial and temporal resolution of PlanetScope imagery allows investigation of the pattern of individual lake filling and drainage events (Figure 5). Analysis of the pattern of individual lake dynamics reveals a complex system, where individual lakes may exhibit seasonal-to-multiannual patterns of surface area variations that greatly differ from the glacier-wide trends and even nearby lakes. We found that larger lakes near glacier termini tend to be more temporally stable than smaller lakes further from the terminus (Figure S11). Many of these larger lakes fill to their annual maximum extent at the end of the monsoon

season and gradually decrease throughout the post monsoon and winter months (e.g. Figure 5b).

In our Planet-derived dataset, we identified multiple large lakes (>10000 m²) which experienced rapid drainage events (Figure 5), which all occur near the onset of the monsoon (May–June). Drainage events similar to these have been previously documented in this region and can cause significant impacts and damage to downstream communities and infrastructure (Rounce et al., 2017; Miles et al., 2018). In fact, the lake which we observed to rapidly drain twice (in 2019 and 2021) was

also noted to be involved in the 2015 flood event documented by Rounce et al. (2017). Further analysis of the Landsat archive





**Figure 8.** The October-February seasonality of SGLs on Lhotse Glacier, with each row showing a unique year. (a) shows the percent of the glacier area with SGL coverage (y-axis) as a function of the normalized distance form the terminus (x-axis). Each line showing the area in a single month. (b) shows the cumulative SGL area (y-axis) with increasing lake size (x-axis), in each month. The x-axis ticks are shown in 900 m$^2$ increments (the size of a single Landsat pixel). Areas are normalized to a 0–1 scale to allow direct comparison between months and years. (c) shows the changes in the number of individual lakes within each image, binned into three different sizes. Blue are smaller in size than half of a Landsat pixel, orange are between one half and two Landsat pixels, and red are greater than two Landsat pixels.





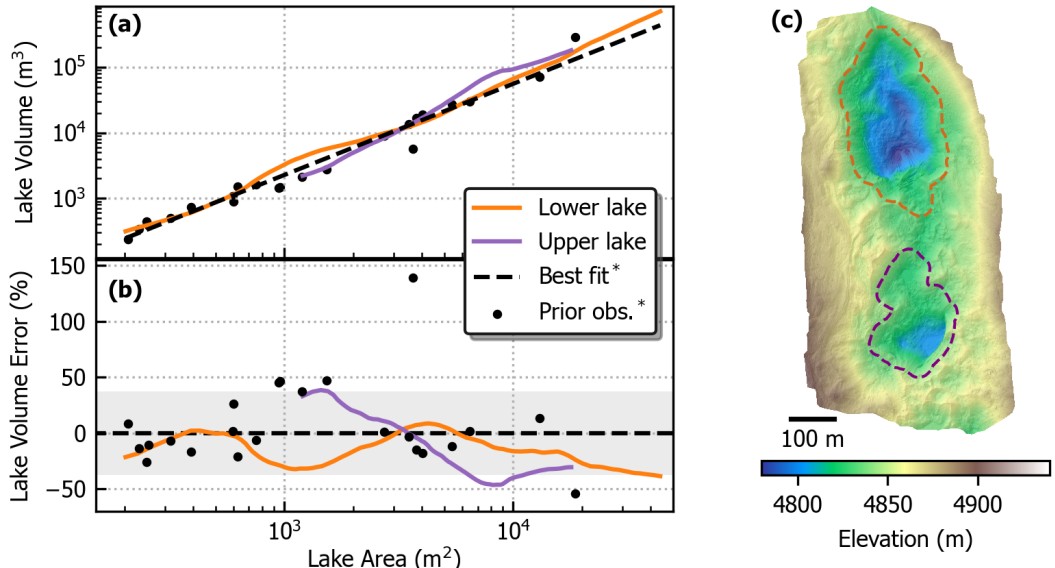

**Figure 9.** The area-volume scaling relationship for the two large lake basins on Ambulapcha Glacier (purple and orange lines). Black points show previous observations of SGL areas and volumes on Khumbu Glacier by Watson et al. (2018b), and the black dashed line is the best-fit relationship presented in the study. (a) shows the area-volume relationship (in log-log scale). (b) shows the percent error that results from using the best-fit equation for estimating volume from an observed area. Grey shaded area shows the 37% mean error from all observations in Watson et al. (2018b). (c) shows the UAS-derived DEM (hillshade with elevation overlaid) and approximate extents of the two lake basins.

reveals six additional instances of this lake filling and draining within the 2008–2016 time period (and none prior to 2008), and the lake area increased each time that it refilled.

The total SGL area on Ambulapcha Glacier is dominated by two large lakes near the terminus that have distinct and contrasting seasonal patterns (Figure 5). The lake nearest to the terminus rapidly fills during the monsoon season each year and slowly

drains throughout the post-monsoon, winter, and pre-monsoon seasons (Figure 5). In contrast, the second lake further from the terminus was consistently present over the duration of the PlanetScope observations, expanded throughout the five-year period, and completely drained during the onset of the monsoon in 2022 (Figure 5).

These lakes were empty during the May 2023 field campaign, allowing detailed topographic mapping the the lake basins (Figure 9). Both lakes seemed to had drained englacially, as evidenced by the presence of small ponds and exposed ice at

the bottom of each basin. We used the high resolution DEMs of the basins to construct an area-volume scaling relationship for each, by calculating the total surface area and volume with incrementally increasing water depths (0.2 m increments) to a maximum depth of 40 and 24 meters (for the lower and upper lake, respectively). The volume of the pre-existing ponded areas in each basin were estimated according to the relationship presented by Watson et al. (2018b), which built on the findings of Cook and Quincey (2015). The results from our artificial filling of the two basins found that the area-volume relationships of

each fall within the range of existing SGL observations (Watson et al., 2018b) (Figure 9).





## 5.3 Long-term trends

Complex patterns and trends in SGL evolution over decadal timescales are revealed by the Landsat-derived products, with notable differences being observed between different glaciers, between different parts of individual glaciers, and through time.

Over the 1988–2022 period, three of the eight glaciers (Khumbu, Lhotse, and Ambulapcha) showed a statistically significant (P<0.05) positive trend in glacier-wide SGL area in Landsat-derived observations, based on a linear regression of the average annual lake area (Figure 10, Figure S11, Table S3). One glacier (Lhotse Shar) showed a statistically significant negative trend, while the remaining four did not have a statistically significant trend. Limiting the time period to only 2013–2022 (after the launch of Landsat 8), we find that only Khumbu has a statistically significant positive trend in SGL area, while all seven other glaciers have no statistically significant trend (Figure 10, Figure S12).

If we limit the area of investigation of each glacier to include only the lower half of the debris-covered area, where lake area shows less seasonal variation, then we find similar temporal trends. Three glaciers (Khumbu, Nuptse, and Ambulapcha) show an increasing trend over the 1988–2022 period, one (Imja) shows a decreasing trend, and the remaining four show no significant trend. In the more recent (2013–2022) period, three glaciers show a statistically significant increasing trend (Khumbu, Lhotse, and Ambulapcha), one (Ama Dablam) shows a decreasing trend (Figure 10, Figure S12, Table S3). The remaining show no statistically significant trend over the 2013–22 period, although lake area on Nuptse has increased while lake area on Lhotse Shar has decreased.

Looking at the decadal-scale changes in spatial distribution of SGLs better illustrates the contrasting patterns among the glaciers studied which we found in the time series trends. The development and expansion of SGLs near the terminus of Khumbu Glacier is clearly visible, while at the same time there is a decrease in the frequency of lakes further up-glacier (Figure 11). Similarly complex patterns of SGL changes are apparent on all glaciers (Figures S13–S19), and offer complementary data to the time series investigation.

## 6 Discussion

### 6.1 Comparison with previous SGL studies

The results of this study highlight how the high temporal resolution of PlanetScope imagery can be leveraged to advance our understanding of supraglacial lake dynamics. Prior analyses of the seasonal cycle of SGLs (Miles et al., 2017; Narama et al., 2017) have largely relied on coarser resolution Landsat imagery aggregated over decadal time periods in order to identify seasonal trends. While our results show a gradual increase in total lake areas throughout the winter leading up to the onset of the monsoon, these analyses (neither of which focus on the Khumbu region) have found that SGLs initially appear and fill in the pre-monsoon months (March to April) and then drain throughout the monsoon. A study by Wendleder et al. (2021) synthesized Sentinel-2, PlanetScope, and SAR datasets to construct a time series of SGL area over the summer months (April–September) on Baltoro Glacier (Pakistan). Similarly to the previous studies, Wendleder et al. (2021) also found that lakes appeared and expanded rapidly in the spring months and drained throughout the summer.





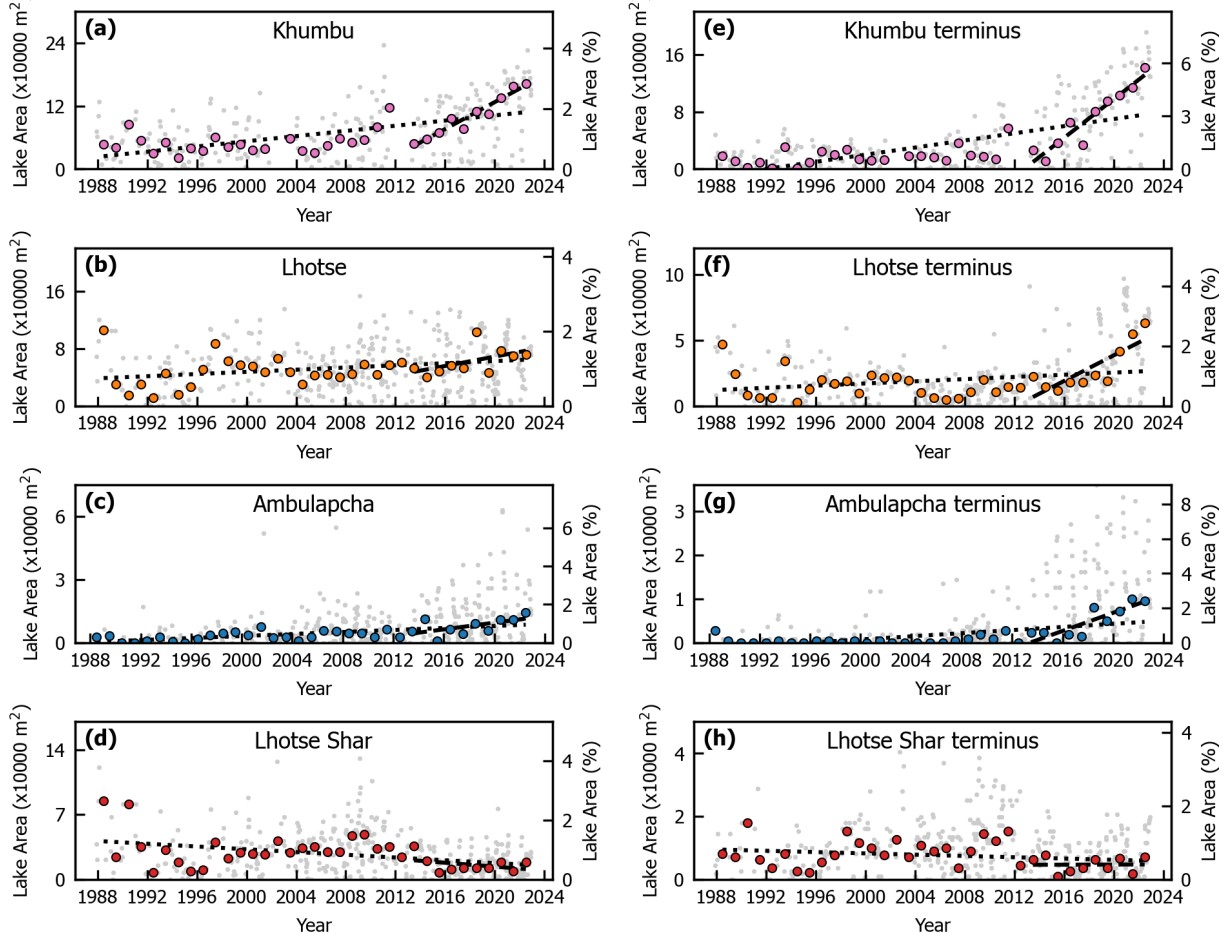

**Figure 10.** Long term trends in SGL area on four glaciers as derived from Landsat imagery. Light grey dots show the lake area in individual LAndsat images, while larger colored dots show the mean SGL area within each calendar year. (a-d) shows the SGL area over the entire glacier surface, while (e-h) show the lake area only on the near-terminus area (defined as the lower 50% of the glacier). Dotted and dashed lines show the linear regression of the annual averages for the entire time period (dotted) and only the 2013–2022 time period (dashed). Slopes and P-values are provided in Table S4.

The differences between these previous studies and our findings showing winter-time lake expansion may be due to method-ological differences, such as coarser-resolution imagery being unable to capture smaller lakes which we observed with Plan-
320  etScope imagery, not including frozen lakes in their inventories, and/or regional variations in SGL filling and draining. The debris-covered areas of the glaciers in our study region generally do not have continuous snow-cover for the entire winter, making SGL identification possible during these months. A study by Taylor et al. (2022) found wintertime increases in SGL




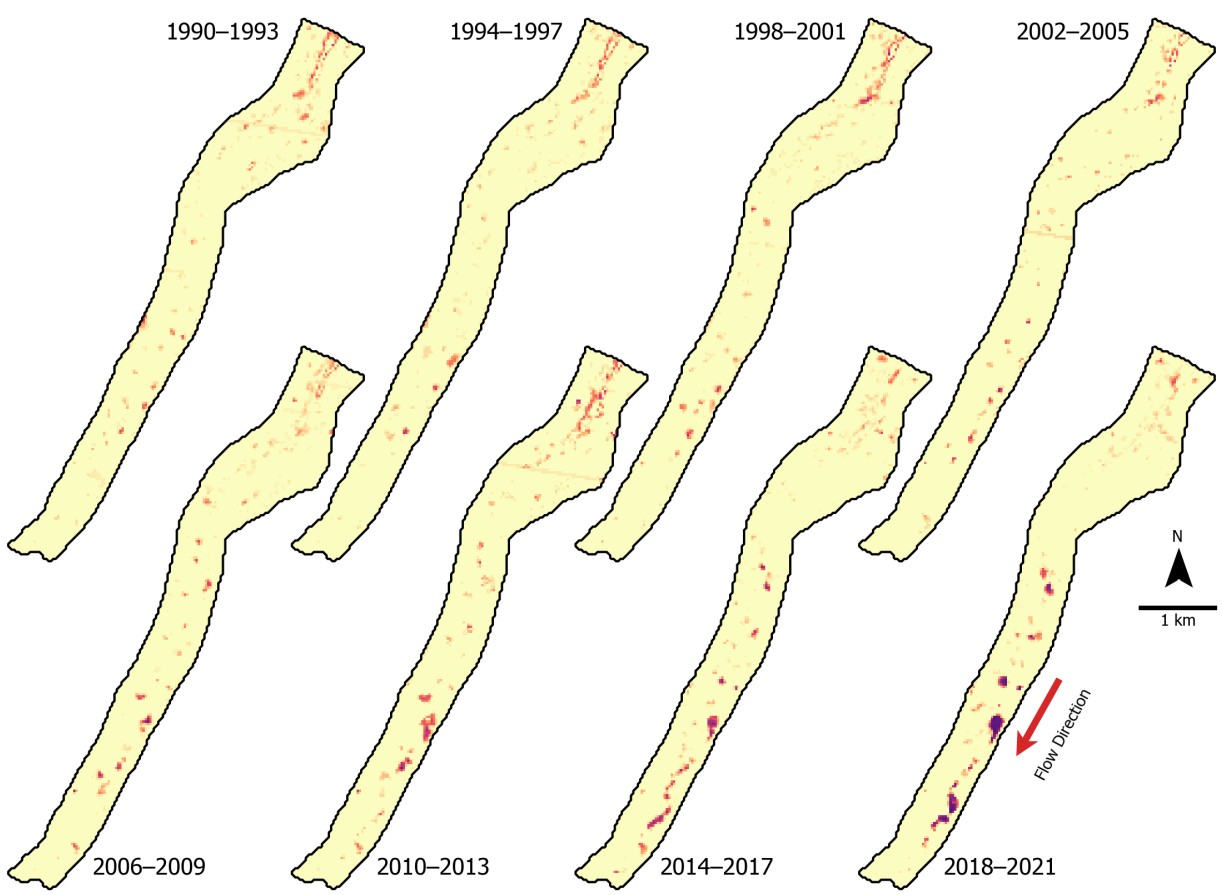

**Figure 11.** SGL frequency maps of Khumbu Glacier as derived from Landsat imagery. Each figure shows the water distribution aggregated within four-year intervals. Colors show the frequency with which each pixel was identified as water, with darker colors indicating more frequent water.

area on two of the three debris-covered glaciers they investigated on the Bhutan-Tibet border, and attributed this to variations in efficiency of supraglacial drainage systems between different glaciers and between seasons.

325     The seasonal variation in SGL area on individual glaciers that we find with our Planet-derived products is of higher magnitude than decadal-scale trends in lake area change from the Landsat record. Many previous studies which use higher-resolution imagery to investigate SGLs on the same glaciers as this study have used a series of high-resolution images that may cover a wide range of years (up to multiple decades) but are captured at irregular seasonal timing (Mohanty and Maiti, 2021; Watson et al., 2016; Steiner et al., 2019). Our results suggest that without controlling for seasonality, such study designs are likely to

330  be insufficient for differentiating between the regular seasonal cycle and long-term evolution of SGLs described here.

    Watson et al. (2018b) presented in-situ observations on SGLs near the terminus of Khumbu glacier over the 2015–2016 year. They found that these lakes reached an annual maximum during the summer monsoon, followed by continuous drainage



throughout the winter months, which largely supported the findings of Miles et al. (2017) and Narama et al. (2017). Our analysis showed a similar annual pattern of lake area in this region of Khumbu Glacier (Figure 5). The discrepancy between the patterns seen in these lakes and total glacier-wide SGL area shows that caution should be applied when extrapolating observations across the entire glacier, to other lakes, or to other glaciers, as individual lakes/portions of the glacier often have unique spatiotemporal patterns

The limited availability of cloud-free imagery in the monsoon months limits our ability to draw conclusions about the specific timing and rate of drainage from our automated SGL identification approach. An approach focused on manual lake delineation during the monsoon and pre-monsoon time periods, or approaches using SAR data (Wendleder et al., 2021) would be more capable of answering these questions.

These findings highlight how, while satellite remote sensing platforms are capable of tracking changes in SGL area, they are not able to identify the specific processes which cause lakes to fill and drain. Detailed in-situ observations such as those by Watson et al. (2018b) are needed to complement remotely-sensed observations.

## 6.2 Winter lake expansion

Our results showing an increase in total SGL area throughout the winter months is seemingly counterintuitive, as the cold temperatures, limited precipitation, and minimal snow and ice melt would seemingly limit the availability of liquid water and suppress lake formation (Miles et al., 2017). We hypothesize that increased absorption of solar radiation and conduction to underlying ice in areas with thin debris cover (Rounce et al., 2021; Miles et al., 2022) may lead to melt water production during winter months, while inefficient drainage systems result in this water pooling at the surface (Miles et al., 2020). This hypothesis is supported by the observation that wintertime lake appearance and expansion is most prominent in the upper debris covered regions of the glaciers, where debris cover is likely thinnest (Figure 8). The inconsistent snow cover on the debris-covered tongues of these glaciers (particularly during the early winter months) would increase the potential solar radiation reaching their surface, particularly on the glaciers with more southerly aspects which also show the most consistent wintertime lake expansion (Figure 7). This phenomenon may not be observed on glaciers in other regions with more consistent wintertime snow cover, or in regions with climates that are not dominated by the summer monsoon.

## 6.3 Long-term trends

Our findings highlight many of the difficulties associated with monitoring the long-term evolution of SGLs. The seasonal variations in glacier-wide lake area are of similar or greater magnitude to the long-term changes (Figure 7, Figure 10). The long-term trends in lake area are variable depending on the time period and the specific area of the glacier investigated. We found that SGL area has increased near the terminus on four of the glaciers investigated (Khumbu, Lhotse, Ambulapcha, and Nuptse) over the 2013–2022 time period. This observed expansion and possible future coalescing of near-terminus lakes on these glaciers suggests that they should be closely monitored for future outburst flood risk (Benn et al., 2012).

An efficient supraglacial stream network on the surface of Lhotse Shar, combined with a highly incised notch in the terminal moraine, as observed from field observations, has likely prevented large lakes from persisting on its surface. In contrast,





Khumbu Glacier has a similarly efficient supraglacial stream network near its terminus, which also flows off the glacier through a deep notch in the terminal moraine, but has nevertheless developed many large SGLs that have been continuously expanding and coalescing in recent years. This contrasting behavior, compared to Lhotse Shar, may be due to a slightly lower surface gradient and greater number of ice cliffs on Khumbu Glacier. However, the presence of the outlet stream deeply incised into
the moraine on Khumbu glacier may help to limit future SGL expansion.

The recent SGL expansion on Lhotse Glacier is unique due to the distance up-glacier at which this expansion is occurring (Figure 8, Figure S17). The lower section of the glacier, downstream from the area of lake expansion, is relatively steep (up to 5° near the terminus), has a well established stream drainage network, and has little-to-no terminal moraine. The lake expansion has occurred predominantly in an area ∼2 km from the terminus, at a point where the surface gradient lowers to 2°
and maintains that slope for ∼3 km up-glacier. The lack of a terminal moraine suggests that the risk of a large terminal lake (similar to Imja Tsho) forming on Lhotse Glacier is low, however sudden drainage of the large SGLs englacially and through the existing supraglacial stream network remains possible (Rounce et al., 2017).

Our long-term time series of SGLs on Imja and Lhotse Shar glaciers show only a portion of the lakes that are present throughout the time period because a static glacier outline was used. These glaciers were significantly larger in the late 1980s,
and have been retreating rapidly due to the expansion of their proglacial lake (Imja Tsho). Their transition to lake-terminating glaciers has likely inhibited the growth of supraglacial lakes due to increasing surface velocities. The expansion of Imja Tsho is certain to have further significant effects and feedbacks on the subglacial, englacial, and supraglacial hydrologic systems of the glacier (Watanabe et al., 2009), but those are not fully investigated in this paper.

### 6.4 Methodological limitations

The automated approaches used for SGL delineation in Landsat, Sentinel, and PlanetScope imagery substantially increases the amount of data which can be efficiently analyzed relative to manual identification methods. However, it is important to understand the limitations of these automated methods in order to appropriately use the derived products.

The geolocation accuracy of PlanetScope imagery can be poor, particularly in complex and steep terrain such as the Himalayas. This can result in multi-pixel misalignment between images, or even between individual bands within the same
image. This can inhibit the ability to track spatial and temporal evolution of features as small as SGLs and introduce substantial noise into the data. Differences in the sensors on individual satellites can further add noise. Taking advantage of the high temporal resolution of PlanetScope imagery allows you to remove much of the noise and gain confidence in the results, but significant cloud cover during the pre-monsoon and monsoon seasons lowers the confidence of the results during those times.

Landsat and Sentinel-2 imagery have better geolocation accuracy, and more consistency in their spectral responses than
PlanetScope imagery, which allows them to overcome many of the aforementioned limitations. However, the lower spatial and temporal resolution (particularly for Landsat) means that smaller SGLs and short-term variations are difficult or impossible to identify. Additionally, identifying lakes with frozen surfaces is more difficult without the dense temporal resolution that was used for the Planet-derived dataset, further obscuring winter-time changes in lake area.



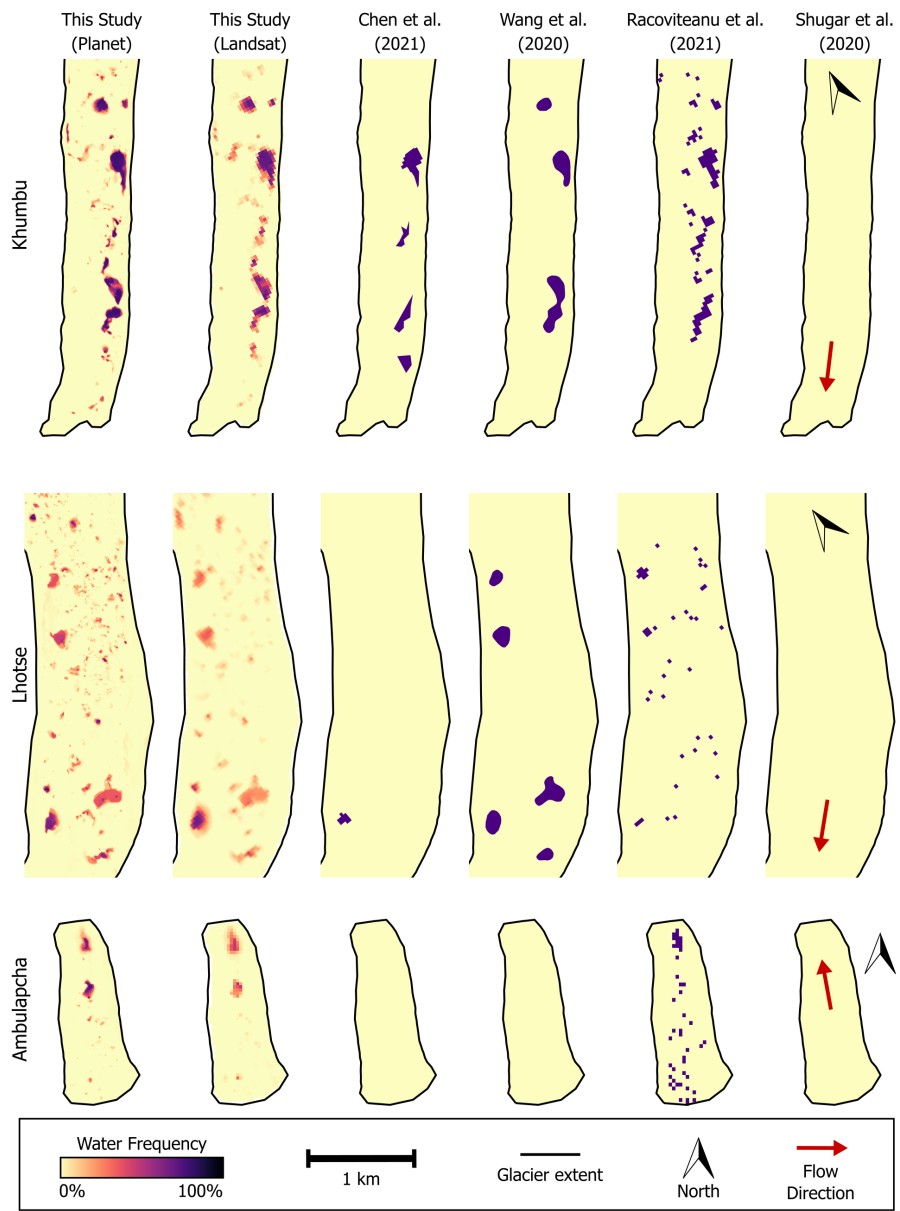

**Figure 12.** A comparison between our Planet-derived and Landsat-derived lake extents and publicly available regional-scale inventories (Chen et al., 2021; Wang et al., 2020; Racoviteanu et al., 2021; Shugar et al., 2020).



## 6.5 Comparison with regional-scale databases

A comparison between the SGL extents produced in this study with regional- or global-scale glacier lake inventories (Chen et al., 2021; Wang et al., 2020; Racoviteanu et al., 2021; Shugar et al., 2020) suggests that existing datasets do not capture the full behavior of SGLs through space and time (Figure 12). We found that three time-varying inventories (Chen et al., 2021; Wang et al., 2020; Shugar et al., 2020) considerably underestimated the extent of modern SGLs on these eight glaciers. The Shugar et al. (2020) dataset does not include any SGLs on these glaciers due to the minimum size threshold of 50000 m$^2$ that

was used, and two areas of glacial ice were misclassified as lakes (not shown on Figure 12 because they are up-glacier from the study region). For the ten years of lake outlines in the Chen et al. (2021) Hi-MAG dataset, a single lake was identified on Lhotse Glacier in one year, seven total lakes on Khumbu Glacier across four years, and no lakes on the other glaciers in our study area. In contrast, Racoviteanu et al. (2021) present a comparable distribution of SGLs to our study, but only provides data for a single time step. In summary, the stark differences between these prior studies and our findings highlight how recent

advances in satellite remote sensing can be leveraged to improve regional- to global-scale glacial lake studies, particularly for SGLs on debris-covered glaciers, given their unique spatial and temporal characteristics.

## 7 Conclusions

In this paper, we presented an automated approach to identifying SGLs on debris-covered glaciers from PlanetScope, Sentinel-2, and Landsat imagery. Lake identification in Landsat and Sentinel-2 was optimized using coincident Planet observations.

These methods were used to map lake extents on eight glaciers in the Khumbu region of Nepal over the 1988–2022 time period.

A regular annual cycle in SGL development was found, with the annual minimum in glacier-wide lake area occurring at the end of the monsoon season followed by a gradual increase throughout the winter season and a maximum in February and March, followed by drainage throughout the monsoon season (Figure 7). The magnitude of seasonal variations were

considerable, with the annual maximum being approximately double the annual minimum. This annual cycle is driven by the appearance and gradual expansion of small lakes in the upper debris-covered regions of these glaciers throughout the winter months, while larger lakes near the glacier termini tended to be more stable or slowly drain during the winter. Over decadal timescales, near-terminus lakes were observed to be expanding and coalescing over on four of the glaciers (Khumbu, Lhotse, Ambulapcha, and Nuptse) (Figure 10, Figure S12), highlighting the need for continued observation to evaluate the evolving

glacier lake outburst flood hazards they present.

The annual cycle in SGL area which we identified has important implications for interpreting prior observations of SGLs. The intra-annual timing of observations must be considered in order to differentiate long-term trends from seasonal variability. A comparison between our results and existing regional-scale glacier lake inventories showed that the complex spatiotemporal patterns are not fully captured in these datasets. Our findings highlight the unique capabilities of satellite remote sensing

platforms to track changes in SGL area over time. However, detailed in-situ observations are needed to complement these remotely-sensed observations to better understand the processes controlling SGL evolution.



*Code and data availability.*   PlanetScope imagery is available through Planet (https://www.planet.com/). Landsat imagery is available through the USGS EarthExplorer platform (https://earthexplorer.usgs.gov/). Sentinel-2 imagery is available through the Copernicus Open Access Hub (https://scihub.copernicus.eu/). Derived products (image-specific lake extents, DEMs, and orthomosaics) and analytical scripts are available

via Zenodo (https://doi.org/10.5281/zenodo.8173102).

*Author contributions.*   Concept and design were done by LZ, DM; programming by LZ; formal analyses by LZ; Fieldwork by LZ, DM, SWM, JJ; DEM generation by JJ, DM; writing by LZ; edit and review by LZ, DM, SWM, JJ; visualization by LZ; funding acquisition and project administration by DM, SWM;

*Competing interests.*   The authors declare that they have no competing interests

*Acknowledgements.*   This effort was supported by NASA's High-mountain Asia research program (NNH15ZDA001N-HMA, award number 80NSSC20K1343). The authors thank Himalayan Research Expeditions for important logistical support during fieldwork.



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
