# Peer review of "Seasonal to decadal dynamics of supraglacial lakes on debris-covered glaciers in the Khumbu Region, Nepal"

_EGUsphere, 2023_

## Author Response (AR1)

Dear Referee,

Thank you for taking the time to review our manuscript and to provide constructive feedback. We feel that the changes made in response will substantially improve the quality and value of the paper. We provide detailed responses to each of your comments below, with our responses in blue.

While formatting our responses, we found a small error in the code used to compute the optimized NDWI threshold for Landsat and Sentinel-2 imagery. The green-band shadow masking applied to Landsat and Sentinel-2 images (lines 164-165 in the initial submission) was mistakenly not applied to the training/validation dataset prior to computing the optimized NDWI threshold for Landsat and Sentinel-2 images. After fixing this omission, the optimized thresholds found are changed slightly (due to the low-brightness shadowed areas being commonly mis-classified as water without this shadow masking applied). These new NDWI thresholds are: 0.137 (Landsat 5 & 7), 0.188 (Landsat 8 & 9), and 0.250 (Sentinel-2), compared to the original values of 0.172, 0.226, and 0.260. Additionally, we introduced an additional filtering step to remove occasional linear artifacts in a handful of Landsat 5 images (see the example below). This filtering step removes pixels where the difference in surface reflectance between green and near-infrared is greater than 0.2. his threshold was chosen to remove the majority of these artifacts while leaving no discernible difference in true water identification.

[Figure]

All analyses and figures which used Landsat and Sentinel-2 data have been updated to reflect these changes. Items which have changed are: Figures 4, 6, 10, 11, and 12 (based on figure

numbering in the original manuscript submission), Supplementary Figures S12-S19, and Tables 1, S3, and S4. The figures below (Figure 6 c-d and Figure 11 a-d) provide an example of the changes which have occurred.

[Figure]

[Figure]

[Figure]

[Figure]

Dear Lucas and co-authors,

your study analyses the seasonal and decadal evolution of the supraglacial lakes on eight neighboring debris-covered glacier in the Khumbu Region, Nepal. Therefore, the satellite images of PlanetScope, Sentinel-2, and Landsat 5-9 were used. The high temporal resolution of PlanetScope enabled the monitoring of the seasonal evolution and Landsat of the decadal evolution since 1988. For the classiciation of the supraglacial lakes, the Normalized-Different Water Index with manually optimized threshold was applied. The study gives an interesting insight into the lake evolution in the Khumbu Region. Unfortunately, the manuscript needs major revisions (see below). The findings must be better elaborated and the results must be specified with exact numbers like lake area or number. If the maximum lake area is End of April, which is differently to other studies, the statement should be underlined with better explanations or evidences and hence to strengthen your key findings. I hope my comments are helpful and I am looking forward to reading the revised manuscript.

All the best,

Anna Wendleder

Major corrections:

The chapter "Methods" is very comprehensive and detailed processing steps like downloading the data or manual check are unnecessary. It would be good to revise this chapter, focus only on the important steps, and shift all important information about the three sensors in the new chapter "Materials".

Thank you for this helpful suggestion, which was similarly brought up by the other reviewer. We agree that the original Methods section was perhaps too comprehensive for The Cryosphere and included information not explicitly needed for interpretation of the findings. In order to focus on the most important steps we have elected to move portions of the Methods section into the Supplementary Materials. Specifically, we have moved the PlanetScope "Data Access and Cleanup" as well as "Filtering and smoothing" sections into the supplement.

What are your key findings regarding the method and the classification results? Please elaborate both points and underline your results more with exact numbers. You derived the supraglacial lakes for 35 years but without mentioning number, maximum area, or date of maximum area. A table with all these parameters would improve your study immense. Furthermore, it would be interesting to analyze why the supraglacial lakes on the eight neighboring glaciers behave differently.

We appreciate this suggestion. We agree that these are important findings to highlight in the context of this project. We have elaborated on these concepts by including a table in the main text that provides information of the number of lakes, lake area, and number of permanent lakes for each glaciers using the PlanetScope product. Additionally, supplementary tables provide this information broken down by individual years. We have expanded the Results section to highlight some of these findings, and have included further discussion of them at various points in the Discussion section.

Reduce the figures - few meaningful figures would underline your core message.
Thank you for this suggestion. We have reduced the numbers of figures in the main text by moving Figures 2, 6, and 9 into the supplementary material. We elected to move Figures 2 and 6 because they are the most methodology-focused, and provided the least value for interpretation of our findings compared to other figures. We moved figure 9 (which investigated the area-volume scaling of the Ambulapcha lakes) to the supplement in order to focus the readers attention on the main findings of the paper (as discussed further below, Line 280).

PlanetScope gives insight into the seasonal evolution and Landsat into the decadal evolution. What is the benefit of using Sentinel-2 in your study? If you would fuse the classification results, then it would make sense to use Sentinel-2 as well. But in this case, the results of Sentinel-2 have no context to the overall results.
Thank you for this clarifying question regarding the benefits of including Sentinel-2 in our study. As you mention, our results and discussion focus primarily on the Landsat- and PlanetScope-derived SGL products due to the different spatial and temporal scales over which they allow investigation (decadal vs seasonal), while Sentinel-2 does not provide novel observations past these (due to having a coarser spatial and temporal resolution than PlanetScope, and over approximately the same time period of 2018-present).

We considered the use of a sensor-fusion approach (which you have shown to be a powerful tool on Baltoro Glacier), but decided against it so that we could compare the effectiveness (and limitations) of each individual sensor. We feel that the inclusion of Sentinel-2 in the methods and results of this study is important part of this, allowing a direct comparison between Sentinel-2 and PlanetScope effectiveness. By including the results from Sentinel-2 (e.g. Figure 4 in the original submission) we provide future investigators insight into how well Sentinel-2 may be capable of capturing seasonal to annual pattern of supraglacial hydrology (relative to PlanetScope) as well as what spatial and temporal patterns may be missed. Additionally, our methodology and results (i.e. the NDWI threshold optimization) provide a simple, flexible approach that could be used in the future to investigate SGLs on a regional scale, something that would be much more difficult to do with PlanetScope imagery.

Minor corrections:

Line 6: Please, indicate a concrete number for the supraglacial lake area and define the period of pre and post-monsoon.
We provide exact numbers for SGL area throughout the main text, figures, and supplementary materials. With the limited word count allowed in the abstract we elected to not include these details here in order to allow more focus on other topics and findings of our study.
Similarly, we define the dates corresponding to the monsoon, post-monsoon, winter, and pre-monsoon seasons in the Introduction section of the main text, and do not include these definitions here in order to save space. We have also gone through the manuscript to ensure consistent use of these terms throughout.

Line 37: Correct to "and Synthetic Aperture Radar".
Done

Line 51: Change to "Therefore, we integrate PlanetScope…".
We have rephrased this paragraph to read: "*The goal of this study is to develop consistent observations of SGL evolution over daily-to-decadal timescales on eight glaciers in the Khumbu region of Nepal through the integration of PlanetScope satellite imagery (Planet Team 2017), Sentinel-2 and Landsat 5–9 imagery, and in situ field observations.*"

Line 63: Do you have a quantity specification for the precipitation?
We have included the average annual precipitation of 587 mm at the Pyramid Weather Station (located adjacent to Khumbu Glacier) as provided in the Perry et al. (2020) publication.

Line 71: What exactly is novel about your approach? Could you define it in 1-2 sentences because applying NDWI is not at all novel? Why are you using the NDWI and which advantages does it has compared to other methods?
Thank you for this clarifying question. The novel aspects of our workflow are: 1) our approach to leveraging the very-high temporal resolution of PlanetScope imagery to to overcome many of the challenges posed by the variable spectral quality of the sensors, and 2) using the semi-automated PlanetScope-derived dataset to build a large validation dataset for coarser resolution imagery. We have modified this sentence to remove the word 'novel' to ensure that we are not mis-using or over-using the term.

Line 73: Change to "observations (3 m spatial resolution)".
Done.

Line 85: Was the exposed glacier ice excluded manually?

Thank you for this clarification. This is correct, when manually delineating the debris-covered areas the areas of exposed glacier ice were excluded manually. We have updated the sentence to make this more clear.

Line 100: It is an important point that you have used PlanetScope data harmonized to Sentinel-2. In case of Landsat, there exist the sen2like data. Have you used them as well?
We did not use any harmonization between Landsat and Sentinel-2 imagery. We elected to use the harmonization tool for Planet images as a way to reduce substantial variations in surface reflectance among PlanetScope images. This processing was intended to create a more self-consistent timeseries of surface reflectances, rather than create a direct match between PlanetScope and Sentinel-2 images. During our testing and analysis, we found that even with this processing step applied there was still substantial variation across PlanetScope images.

Line 109: Could this limited in time, for example during winter (November to March)?
Due to the vertical relief in this region, there were some sections of glaciers which were obscured by terrain shadows for the entire calendar year, although the effects were much greater in the November to March time period.

Line 116 ff.: How were the thresholds (0.10, -0.15, and 0.02) defined? Empirically?
These thresholds were determined empirically through testing of many values. We have updated the manuscript to include this detail.

Line 119: Would the red band be better for shadow detection compared to the RGB?
We elected to use the average of the red, green, and blue bands in order to avoid issues with missing data and anomalously high or low spectral reflectance of individual bands in some images.

Line 121: What is the advantage of using DN here in comparison with the surface reflectance?
We used the DN here based on initial empirical testing of the methods and thresholds that found more consistency in ice/snow delineation with the DN than with the scaled reflectance values, possibly due to issues with sensor-specific spectral quality in very dark or very bright areas.

Line 165: How were the thresholds defined?
These thresholds were determined empirically through testing of multiple values. We have updated the manuscript to include this detail.

Line 179: Please delete the sentence "calculated after scaling images to surface reflectance" as optical data are mostly
Done.

Line 198: Please write out the abbreviation.
Done.

Line 199: "…0.04 m pixel spacing."
Done.

Line 211: What do you mean with physically realistic?
By 'physically realistic' we meant that the lake locations, sizes, and shapes were what was expected based on manual inspection of the images. This is meant as a qualitative judgement of the products that is then confirmed with the quantitative investigations.

Line 218: Please indicate a concrete number of your results. Have you used the same acquisition time as Watson et al. (2016?).
Thank you for this clarifying comment. There was no direct overlap in the years studied between our two studies (with differences of up to 20 years in some cases), so it is not possible to provide a direct comparison of results. However, we have included the average percent difference between their areas and our observations that are within 15 days (17.3%).

Line 220: How much was these large differences?
We have removed this sentence comparing our findings on Khumbu Glacier as it was mistakenly included from a previous draft based on preliminary results. The differences in lake area is no longer as large as the preliminary results suggested.

Line 263: How do you define larger and smaller lakes and do you have a concrete number of how many lakes are stable?
Thank you for this comment. We have expanded the results presented here to provide additional information on lake sizes, frequency of appearance, and permanence, along with tables (in the main text and supplement) summarizing the glacier-specific lake numbers and areas.

Line 264: At which distance from the terminus? How large is the annual maximum extent?
See above.

Line 280: Is the high-resolution DEM acquired by UAV or do you refer to the NASA DEM? I am very sorry, but I do not see the link between the derived area-volume relationship and the seasonal evolution of the lakes. It does not fit to the context. What was your intention here?
Thank you for pointing out this apparent disconnect between the area-volume relationship and seasonal evolution of lake areas in our submission. As we outline below, we feel that it is a valuable contribution to our collective understanding of the hydrology of these debris-covered glacier systems, but needs to better justified and connected than we have done previously. Making observations of supraglacial lake area is a relatively straightforward undertaking from

remote sensing observations, and is thus the primary tool used for studies such as this one. However, understanding the volume of water stored in these features if often the end goal, as the volume storage (and change) is more important from the perspective of understanding hydrologic fluxes and potential hazards. The use of area-volume scaling relationships is typically used to convert from observed area to estimated volume of lakes (Watson et al., 2018; Cook and Quincey, 2015), but these relationships are far from perfect and can result in large errors. Further, the area-volume relationship for supraglacial lakes/ponds of the size seen on the glaciers in this study ($<20000$ m$^2$) is based almost entirely on the dataset from Watson et al. (2018) of bathymetry measurements of 24 lakes on Khumbu Glacier. Very few other measurements of SGL bathymetry/geometry exist in this region, in part because the effort required to gain these measurements is so large. The observations which we provide in this study of the area-volume relationship of two dynamic SGLs on Ambulapcha Glacier are an important addition to this sparse dataset, particularly the finding that the geometry of these lakes fit within the range of existing observations.

Given this context, we feel that this data is a valuable component of this manuscript. We have elected to move Figure 8 (illustrating the area-volume relationship of these lakes) to the supplementary material as we feel that the other results and discussion presented in the manuscript are more important for readers to focus on.

Line 290: Could you indicate the slope of the positive trend?
The slopes of each linear regression are provided in the supplementary material (Table S8). Additionally, we have edited the figure to include the slopes of lines for the 2013-2022 time periods.

Line 297 ff: Could you indicate exact numbers for the trends? Do you have an explanation for the significant increasing trend at Khumbu, Lhotse, and Ambulapcha – it would be really interesting to understand this phenomenon.
The slopes of each linear regression are provided in the supplementary material (Table S8). Additionally, we have edited the figure to include the slopes of lines for the 2013-2022 time periods.
Regarding the explanation for increasing trends on these glaciers, there is no simple and obvious explanation as to why these glaciers (and not others) are showing increasing lake area. We briefly touch on this in the Discussion section ("Long-term trends"), however a more thorough investigation of the physical processes driving these observations is beyond the scope of the paper.

Line 318 ff: Are the differences mainly due to a different seasonal evolution? The explanation of the coarser resolution does not agree with my study as I used Planet Scope with a spatial resolution of 3 m as well (the classification result was resampled to 10 m afterwards). I think that

the different seasonal evolution is more due to climatic conditions. Have you checked precipitation and temperature of your study site? I am also wondering how lakes in winter could be detected when they are snow covered.

Thank you for this clarifying question. The original wording of this sentence ("regional variations in SGL filling and draining") was confusing and did not capture the point which we were trying to make. The differences in local climate and meteorology are sure to result in different seasonal filling and draining patterns of SGLs, particularly across distances as large as that between the Khumbu and Baltoro regions. We have clarified this point, and have re-ordered the list to reflect that this is likely to be a major factor when comparing SGLs in different regions. Future studies that investigate seasonality in SGLs across larger regions (rather than on individual glaciers or small catchments) with self-consistent lake identification methods would be more capable of identifying the climatic drivers.

In regards to the identification of snow-covered lakes, this was possible in our study only because the glacier surfaces were very rarely completely snow covered during our period of investigation, allowing us to assume that lakes that transitioned directly from open water to snow or ice were in fact still lakes, rather than snow drifts. I do not think for glaciers with complete snow cover all winter it would be possible to differentiate snow-covered lakes.

Line 319: Please specify the definition of smaller lakes.
We have rephrased this line to:
*The differences between these previous studies and our findings showing winter-time lake expansion may be due to methodological differences (such as minimum lake size identifiable in Landsat images being 225 $m^2$ if the image is pansharpened)*

Line 322: Could you please define more precisely which months you are talking about here?
We have edited the text to include the months (December to February).

Line 326: Repetition in one sentence ("higher-resolution imagery" / "high-resolution images").
Thank you for pointing this out. We have fixed the wording.

Line 329: What do you mean with "without controlling for seasonality"?
We have edited this sentence to better explain this, by adding "(i.e. using only images captured during the same month across all years)".

Line 333: How does the same pattern look like?
We have expanded this sentence to better describe the similar patterns of SGL area that were found in the three papers being discussed here. All three found that the SGLs they studied reached their annual maximum area during the monsoon and decreased during the following post-monsoon and winter months.

Line 334: Could you please explain the discrepancy in detail?
Thank you for this clarifying question. This sentence was meant to compare/contrast the annual patterns of lake filling and draining of these handful of lakes near the terminus of Khumbu (which have a maximum observed area in the post-monsoon/September and drain throughout the following months) with the annual pattern of glacier-wide SGL area which shows continuous increases in area throughout the post-monsoon and winter months. We have updated the paragraph to better explain this.

Line 361: How much increased the SGLs?
We have added figure and table citations to the end of this statement on lake expansion on these glaciers in order to clarify the values which we are referring to.

Line 385: If you are defining the thresholds empirically, I would not use the wording "automated approach" as it is more a semi-automatic approach.
Thank you for this suggestion, we agree and have edited the wording accordingly.

Line 388: Could you specify "poor" in numbers?
We have updated this sentence to include geolocation accuracy estimation provided by Planet.

Line 392: Please change to "… imagery allows to remove …"
Thank you for this suggestion. We have edited this sentence to read "…allows the filtering out of…"

Line 392: How large was the noise? Please be more exactly with your statement.
The amount of 'noise' is dependent on the specific approaches used and the goal of an analysis. The goal of this section is to provide a general statement on how the high temporal resolution of PlanetScope imagery is capable of overcoming some of the limitations of the PlanetScope spectral quality and geolocation accuracy, rather than to provide insight into the precise signal-to-noise ratio that we run in to for this application.

Line 394: Do you have an accuracy for the better geolocation?
Similar to the comment above, this statement is meant more as an overarching observation of the comparison between using Landsat, Sentinel-2, and PlanetScope imagery in this region and is based on personal observations of the image quality. We have updated the phrasing of the paragraph to better reflect this purpose.

Line 397: Perhaps it is possible to identify ice with Sentinel-2 or Landsat using other bands than visible and NIR?

The use of additional bands in Landsat and Sentinel-2 (particularly SWIR and land surface temp) allows easier identification of ice and snow. However the differentiation between lake ice and glacier ice or drifted snow would be difficult without the very high temporal frequency that PlanetScope images provides and which we utilized to infer what ice/snow surfaces were actually frozen lakes.

Line 400 ff: Please describe more the other studies (which data have been used, which methods, acquisition date of the satellite images). For a better comparison, it would be better to indicate number and lake area of the other studies.

Thank you for this suggestion. We have expanded this section of the discussion to provide a more clear comparison of the goals and findings of each of these studies, and to give a more fair comparison between their results and ours.

Line 420: Please, indicate maximum and minimum area in exact numbers.

We have included these annual maximum and minimum areas for each glacier in each year (from PlanetScope imagery) in the Supplementary material.

Figure 1: Please add the coordinate frame. If the illustrations are aligned to the North, there is no need for a north arrow anymore.

Thank you for this suggestion, we have edited the figure accordingly.

Figure 2: Your workflow is divided into different boxes which are indicated with a), b), etc. Perhaps it would be better to use the chapter number to make it easier to read? Please, include the intermediate steps and the queries of the thresholds as well into the workflow.

We agree that this is a better approach to dividing the workflow to make it easier to follow. This figure has been moved to the supplementary material to allow greater focus on the primary results in the main text. Further edits to the methodology section have removed the intermediate chapter/section numbers. As such, we have elected to not make further edits to this figure.

Figure 3: Please add the coordinate frame and a white glacier boundary would be better visible.

Thank you for this suggestion, we have made the glacier outline white. The images in this figure have been rotated, making a coordinate frame on the image edges not feasible (without adding an overlaying grid, which causes the figures to be overly complex).

Figure 4 upper part: Please indicate a legend and a coordinate frame. A north arrow is not needed here. It would be better to enlarge the classification results and to shift them together I assume that the colors indicate the water frequency – do you think that the higher temporal resolution of PlanetScope leads to a higher water frequency?

Thank you for these suggestions. We have added a legend (colorbar) for the upper images and enlarged them. We do not feel that a coordinate frame is needed in this context given that the

location of the glacier is shown in the study area figure (Figure 1). The higher water frequency observed in the PlanetScope product is likely due to the inclusion of ice and snow-covered lakes in this dataset.

Figure 4 lower part: Please add more x-tic (every 3 months) and indicate the monsoon period in greyish (c.f. Figure 5). If you shift the legend into the diagram, you could enlarge the diagram to make it easier to read. How do you explain the differences between the sensors?
Thank you for these suggestions. We have added additional x-axis tick marks to the time series. We tested variations of the legend size and location, including placing it within the diagram frame, but found that this layout was the most readable and clear. The differences between the sensors are likely due to a combination of different methodologies for lake identification (between PlanetScope vs. Landsat and Sentinel-2), and variable spatial resolution of the sensors.

Figure 5: It would make sense to put all your results (number and area of the lakes) into a table to better compare the lakes on the different glaciers.
Thank you for this suggestion. We have added tables in the main text and supplement which provide the range of lake number and area for each glacier in each year.

Figure 6: Do you mean vertical instead of horizontal lines? Please correct to "with each point representing…".
Thank you for pointing out these typos, we have fixed them in the updated manuscript.

Figure 7: Why did you only use PlanetScope when you have all three sensors available? Please indicate the period of the monsoon.
We have included the dates on the monsoon in the figure caption. We elected to only include PlanetScope data for these plots because it is the dataset which best captures the regular seasonal evolution of SGL area, and there is insufficient space to include data from multiple sensors.

Figure 10: Correct to "…individual Landsat images,…". Could you specify the slope of the linear regression in the diagram?
Thank you for this suggestion. We have updated these figures to include the slope of the 2013-2022 linear regression in each plot, and the slopes for all lines on all glaciers in provided in the Supplementary Material (Table S9)

Figure 11: Does the frequency depend on the frequency of the acquisitions as well? I am sure, there were not that many acquisitions in the early 1990ies as it is nowadays. Please add the legend.
Thank you for these clarify comments/suggestions. We have added a legend to indicate the frequency. The frequency shown is independent of the number of images, as the colors are normalized by the number of observations.

Table 1: If PlanetScope data are harmonized to Sentinel-2, shouldn't they have the same threshold? I would delete the table as you indicated the threshold in Figure 5 and in the text.

Thank you for this clarifying question. The harmonization tool used for PlanetScope imagery (the processing is done by Planet before the imagery is delivered) was used to create a more self-consistent timeseries of surface reflectances, rather than create a direct match between PlanetScope and Sentinel-2 images. During our testing and analysis, we found that even with this processing step applied there was still substantial variation across PlanetScope images and differences between the PlanetScope imagery and individual Sentinel imagery. Additionally, the differences in methodology used for lake identification in the PlanetScope and Sentinel-2 imagery (i.e. the use of spatially-varying thresholds, temporal smoothing of PlanetScope data, and inclusion of ice-covered lakes) results in different thresholds.

We have elected to keep this table in the text (to allow easier identification of the thresholds for readers) and have moved Figure 5 to the supplement in order to focus more on the results of the study rather than the methodology.

Dear Referee,

Thank you for taking the time to review our manuscript and to provide constructive feedback. We feel that the changes made in response will substantially improve the quality and value of the paper. We provide detailed responses to each of your comments below, with our responses in blue.

While formatting our responses, we found a small error in the code used to compute the optimized NDWI threshold for Landsat and Sentinel-2 imagery. The green-band shadow masking applied to Landsat and Sentinel-2 images (lines 164-165 in the initial submission) was mistakenly not applied to the training/validation dataset prior to computing the optimized NDWI threshold for Landsat and Sentinel-2 images. After fixing this omission, the optimized thresholds found are changed slightly (due to the low-brightness shadowed areas being commonly mis-classified as water without this shadow masking applied). These new NDWI thresholds are: 0.137 (Landsat 5 & 7), 0.188 (Landsat 8 & 9), and 0.250 (Sentinel-2), compared to the original values of 0.172, 0.226, and 0.260. Additionally, we introduced an additional filtering step to remove occasional linear artifacts in a handful of Landsat 5 images (see the example below). This filtering step removes pixels where the difference in surface reflectance between green and near-infrared is greater than 0.2. his threshold was chosen to remove the majority of these artifacts while leaving no discernible difference in true water identification.

[Figure]

All analyses and figures which used Landsat and Sentinel-2 data have been updated to reflect these changes. Items which have changed are: Figures 4, 6, 10, 11, and 12 (based on figure

numbering in the original manuscript submission), Supplementary Figures S12-S19, and Tables 1, S3, and S4. The figures below (Figure 6 c-d and Figure 11 a-d) provide an example of the changes which have occurred.

Before

[Figure]

After

[Figure]

Before

[Figure]

After

[Figure]

This study brings together a range of satellite image sources to characterise the dynamics of surface lakes (or ponds) on debris-covered glaciers in the Everest-region of Nepal. The analysis is neatly divided between looking at the seasonal dynamics of these ponds using daily Planet imagery, and the long-term (decadal) patterns captured in the Sentinel and Landsat archives. The result is a considerable dataset that builds on previous work on the same glaciers and adds an element of detail with the availability of the finer-resolution imagery. I support its publication, but I do think more could be made of the analysis/data presentation so that future studies, which look to build on this one in coming years, can readily use it as a baseline for comparison.
In particular, the authors should consider:

1. Adding summary statistics to the main text (or, if you prefer, in a table). What are the mean, max and min lake areas for example, per glacier per year, as well as mean, max and min number of lakes? Which glacier hosts the most lakes/greatest area? Is that the same every year? How many lakes are ephemeral vs permanent? How much of the overall lake area do the ephemeral lakes account for (and therefore how important are they, relatively speaking?). These sorts of stats help the reader to interpret the patterns you talk about in the text in general terms, as well as providing concrete values for future studies to use as comparison.

We appreciate this suggestion, which was similarly brought up by the second reviewer. We agree that these are important findings to highlight in the context of this project. We have elaborated on these concepts by including a table in the main text that provides information of the number of lakes, lake area, and number of permanent lakes for each glaciers using the PlanetScope product. Additionally, supplementary tables provide this information broken down by individual years. We have expanded the Results section to highlight some of these findings, and have included further discussion of them at various points in the Discussion section.

2. Providing more information on the life-cycle of these smaller lakes that appear to be responsible for the seasonal patterns you show. For example, and since you have already gone to the trouble of correcting for surface displacement, can you elaborate on how frequently lakes appear and then drain, how long they last (more or less than a single season?), how often they coalesce, and whether it is the same ones that reappear each time, or new ones that emerge? Kneib et al., 2021 do a nice job of this for ice cliffs as an example. This will tell us more about the processes that are driving the surface changes on these glaciers and add significant value to your manuscript.

Thank you for this suggestion. Along with the glacier-specific lake number and area which we discussed in the comment above, we have added statistics which investigate the permanence and recurrence frequency of lakes using the PlanetScope dataset. This line of investigation has

provided important context to highlight and contextualize the wide range of behavior which these dynamic features can show.

3. Whether the inclusion of the UAV data is necessary – I'm not sure at present it adds anything to the key story – if anything it detracts from it. Consider re-packaging it as a ground validation dataset for the Planet imagery (see below)?

We appreciate your comments and suggestions regarding the use of UAV-derived datasets here. We agree that using the UAV-derived imagery as a ground truth for the coarser-resolution imagery would be a valuable approach. Unfortunately, the Planet-derived dataset (2017-2022) does not overlap with the field observations (May 2023), and extending the timeline of the Planet-derived dataset for this purpose is not feasible due to the quota limits and effort which would be needed.

Thank you for pointing out this apparent disconnect between the area-volume relationship and seasonal evolution of lake areas in our submission. As we outline below, we feel that it is a valuable contribution to our collective understanding of the hydrology of these debris-covered glacier systems, but needs to better justified and connected than we have done previously. Making observations of supraglacial lake area is a relatively straightforward undertaking from remote sensing observations, and is thus the primary tool used for studies such as this one. However, understanding the volume of water stored in these features if often the end goal, as the volume storage (and change) is more important from the perspective of understanding hydrologic fluxes and potential hazards. The use of area-volume scaling relationships is typically used to convert from observed area to estimated volume of lakes (Watson et al., 2018; Cook and Quincey, 2015), but these relationships are far from perfect and can result in large errors. Further, the area-volume relationship for supraglacial lakes/ponds of the size seen on the glaciers in this study ($<20000$ m$^2$) is based almost entirely on the dataset from Watson et al. (2018) of bathymetry measurements of 24 lakes on Khumbu Glacier. Very few other measurements of SGL bathymetry/geometry exist in this region, in part because the effort required to gain these measurements is so large. The observations which we provide in this study of the area-volume relationship of two dynamic SGLs on Ambulapcha Glacier are an important addition to this sparse dataset, particularly the finding that the geometry of these lakes fit within the range of existing observations.

Given this context, we feel that this data is a valuable component of this manuscript. We have elected to move Figure 8 (illustrating the area-volume relationship of these lakes) to the supplementary material as we feel that the other results and discussion presented in the manuscript are more important for readers to focus on.

4. Reducing the number of figures overall (including in Supplementary) and condensing the text where possible (particularly methods)

Thank you for this helpful suggestion, which was similarly brought up by the second reviewer (regarding limiting the number of figures and text in the main text). We agree that the original Methods section was overly comprehensive and included information not explicitly needed for interpretation of the findings. In order to focus on the most important steps we have elected to move portions of the Methods section into the Supplementary Materials. Specifically, we have moved the PlanetScope "Data Access and Cleanup" as well as "Filtering and smoothing" sections into the supplement.

Additionally, we have reduced the numbers of figures in the main text by moving Figures 2, 6, and 9 into the supplementary material. We elected to move Figures 2 and 6 because they are the most methodology-focused, and provided the least value for interpretation of our findings compared to other figures. As discussed in our response to your comment on inclusion of UAV-data, we moved Figure 9 (which investigated the area-volume scaling of the Ambulapcha lakes) to the supplement as well.

While we acknowledge that there are many figures provided in the supplementary material, we feel that these are all important to include in order to be transparent and thorough in our presentation of our findings. The majority of supplementary figures are to provide glacier-specific recreations of Figures 8 and 11 (from the original figure numbering). Including Fig. S3-S10 in necessary to highlight how the seasonal pattern of lake expansion during the winter, which we identified from the PlanetScope-derived SGL dataset (one of the main findings of this study), is repeated on almost all glaciers in all years, and is not just found on Lhotse Glacier. The inclusion of the Landsat-derived decadal changes in SGLs on each glacier (Fig. S13-S19) is important because this a relatively data-sparse region, and it could provide important context to future researchers investigating these glaciers either from remote sensing observations or planning field work in this region.

5. Being more explicit about the % errors on your lake areas, rather than presenting it as a proportion of the debris-covered area.

Thank you for this suggestion. We agree that it is important to be explicit in the error estimations, and that presenting errors as both a percentage of the total lake area and the total debris-covered area is important. In order to present our findings in a similar manner to previous studies we have amended the text throughout to include these numbers, including recalculating error estimates for Table S3 (comparing PlanetScope-derived SGL area to Landsat and Sentinel-2 derived areas for coincident imagery).

More minor comments:

Line 11: I interpret 'annual' variation to mean from one year to the next, but I think you're meaning from one season to the next here?
Thank you for this clarifying comment. This was our intention and we have revised it accordingly.

Line 20: present -> presents
Done.

Line 50: can you add a sentence here to underline the importance of understanding these seasonal cycles that your paper characterises?
Thank you for this suggestion. We have elaborated on this concept in the revised manuscript.

Line 52: I'm not sure if it is 'in-situ' or 'in situ' but be consistent
Thank you for pointing this out. We have changed all instances to the correct "in situ".

Line 66 (Section 3): this is a very long section! Can you be a bit more concise and focus the text on key points?
We have shortened the methods section by moving some information in to the supplementary material, leaving only the most important aspects of the methodology in the main text.

Line 68-69: remove sentence starting 'Each individual source…' – it's superfluous
We have changed this sentence to read: "Each source allows…"

Line 75: Unmanned -> Uncrewed
Done

Line 75-76: if that's what your UAV data helped with, I'm not sure it is evident in this paper? Consider revising?
Thank you for this suggestion. We have rephrased this sentence to better reflect the use of in situ observations to provide context of smaller-scale surface features and local topography of the glaciers.

Line 114: maybe spell out the NDWI using mathematical notation?
Done.

Line 116-117: does this methodological step not fall down for larger lakes given all pixels within that 150 m buffer will have similar NDWI values? Can you clarify?
Thank you for this clarifying question. In our testing, none of the lakes in our study area are big enough to see this effect (Figure 5 highlights the largest lakes we encounter). This length was tuned empirically to be as small as possible while not seeing this effect on the large lakes.

However, if this method was used on larger lakes (for example, Imja Tsho) a larger buffer size would be necessary.

Line 190 (Section 3.3): I'm not convinced that the inclusion of these field data adds value to the manuscript. How about using them instead to assess the uncertainty in using your PlanetScope imagery as the 'truth' for the other coarser resolution datasets? At 3 m spatial resolution there is still some ambiguity as to exactly where the lake margins lie – and with your drone imagery you can put a figure on that, which will be of use to anyone using Planet as a validation dataset going forward?

Thank you for this suggestion regarding the best use of our field observations. We agree that using the UAV-derived imagery as a ground truth for the coarser-resolution imagery would be a valuable approach. Unfortunately, the Planet-derived dataset (2017-2022) does not overlap with the field observations (May 2023), and extending the timeline of the Planet-derived dataset for this purpose is not feasible due to the quota limits and effort which would be needed. Please see our response to the major comments (above) for more discussion on the inclusion of these materials.

Line 200: can you be sure that with an error estimate of +/- 1 pixel these manual delineations are more robust than the semi-automatic results? Normally one would use a finer-resolution dataset than that being evaluated to produce these values...? See point above.

We believe that the manual delineations of SGLs in Planet imagery are more robust than any individual lake product from the semi-automated approach. If we were to manually delineate the lakes on every Planet image we have, then we would have higher confidence in the results. But the time required to do so is of course prohibitive. We consider our validation dataset more as a guideline of how well the semi-automated results match up with that "best possible" dataset of all manually-derived products, and a confirmation that the results and seasonal variations that we find from the semi-automated results are in fact real.

Line 213: can you also state the % error on the lake areas? It's useful to know within the context of the whole debris-covered area, but probably more important is what it means for the data you present in the plots (and ideally, these would have error bars on too)

Thank you for this suggestion. We have now included the % error relative to total lake area. In the text (25%, 33%, and 13% of the total lake-covered area on each glacier, dropping to 7.6%, 19.1%, and 8.3% during the September-February months).

Line 228-230: same point as immediately above

This stated error is calculated with respect to the total lake area. We have updated the text to make this clear.

Line 255-256: this same point has been made three times in quick succession. Maybe remove this sentence?
Thank you for pointing this out. We have edited this section to make it less redundant.

Line 278-285: this feels a bit odd in the context of the satellite-based observations.
Please see our response to the major comments point #3 above.

Line 309-317: Not discussing your results in comparison to Watson et al., 2016, who used similarly fine-resolution imagery along with Landsat to look at seasonal patterns, seems like a bit of an obvious omission here?
Thank you for pointing out the unintentional omission here. We have added a paragraph here to elaborate on direct comparisons between our PlanetScope-derived results with their high-resolution imagery.

Figure 11: needs a colour scale to show frequency values
Thank you for pointing this out, we have included a colorbar in the revised version.

Line 348: can you explain (in the text) why there would be increased solar radiation absorption during winter months?
The use of "increased" was meant to refer to increased energy absorption in the areas of thin debris cover relative to clean-ice surfaces due to decreased albedo, however this was not clear in the original phrasing. We have edited this sentence by removing the word 'increased' here, and further clarifying the explanation in the following sentence.

Line 371: I agree the topographic characteristics of Lhotse are unique within this suite of glaciers, but is the predominant expansion of ponds at Khumbu not also a couple of km from the glacier terminus?
Thank you for pointing this out. Yes, the distance-from-terminus at which the major lake expansion is happening is similar on both Khumbu and Lhotse (and also similar to where SGL expansion initiated on Imja Glacier in the mid-1900s). The down-glacier characteristics (the steepness and lack of a terminal moraine) are the more unique aspect. We have edited this paragraph to put more emphasis on this.

Line 372: Figure S17 takes me to Lhotse Shar, not Lhotse
Thank you for pointing this out. We have fixed the citations and ordering of supplementary materials.

Line 399 (Section 6.5): I'm not sure this is packaged up in a fair way – the other papers/inventories you refer to here didn't attempt to delineate small ponds on the glacier surfaces as you have here – they all set a minimum pond area for detection, and were largely

focussing on what may otherwise be termed as a lake (i.e. much larger than a pond) because of their much broader spatial coverage. It's a bit like comparing apples with oranges in my mind.

Thank you for this suggestion. We have expanded this section of the discussion to provide a more thorough explanation of the previous studies which we are comparing our results to, as well as explaining more clearly that the minimum size threshold used in these studies is the reason for the differences. We feel that it is important to provide the comparisons here in order to highlight the limitations of our current knowledge of the regional-scale spatial and temporal variability in supraglacial hydrology of debris-covered glacial systems.

Figure S12: is it worth pointing our somewhere that the negative trend you identify on Lhotse Shar and Imja is at least partly a consequence of the glacier area shrinking (and Imja Tsho expanding) over the period of observation?

We do not incorporate time-varying glacier extents into our analysis. The products we present here are derived using a single outline corresponding to the modern ~2022 extent. which excludes Imja Tsho. The long-term negative trends for Lhotse Shar and Imja show the lake area for this 'patch' that would have been further removed from the terminus and proglacial lake in the 1980s and 90s. We discuss the implications of this, and the possible feedbacks between proglacial lake expansion and supraglacial hydrology, in the discussion (Section 6.3. Long Term Trends).

Figures S13-S19: these all need a legend to give meaning to the shades of red.

Thank you for pointing this out. We have updated the figures to include the legend.

Figures S13-S19: There are some suspicious areas in the uppermost part of your debris-cover boundary that look to be misclassification, rather than lakes. They are particularly apparent in the Lhotse Nup (2002-2005), Ama Dablam (1998-2001) and Ambulapcha (1998-2001) figures – could they be areas of wet snow? Do they translate through to your data presentation in the main text? Or do you believe them to be genuine…?

These areas that you have pointed out are most likely misclassified shadows, wet snow, or shadowed snow areas that were not masked out during our processing steps. While they are not genuine, these over-estimations of lake area have minimal effect on the long-term trends which we observe because they occur infrequently.

References

Kneib, M., Miles, E. S., Buri, P., Molnar, P., McCarthy, M., Fugger, S., & Pellicciotti, F. (2021). Interannual dynamics of ice cliff populations on debris-covered glaciers from remote sensing observations and stochastic modeling. Journal of Geophysical Research: Earth Surface, 126(10), e2021JF006179. https://doi.org/10.1029/2021JF006179

Watson, C. S., Quincey, D. J., Carrivick, J. L., & Smith, M.W. (2016). The dynamics of supraglacial ponds in the Everest region, central Himalaya. Global and Planetary Change, 142, 14–27. https://doi.org/10.1016/j.gloplacha.2016.04.008

---

## Author Response (AR2)

**Dear Referee,**

**Thank you for taking the time to review our revised manuscript. We have provided point-by-point response to your comments below.**

Dear Lucas,

Thanks a lot for the revised manuscript. It has improved significantly. Great work!

I just a few, small corrections which you could add:

Line 389: "of the of maximum SGL area" to "of the maximum SGL area"
**Done**

Line 477/478: delete blank before comma "(Planet Team, 2022) , particularly"
**This space has been removed.**

Table 2: Please refer the observation period and it would be nice to know the ratio between glacier area and slg area to better compare the different study sites.
**Thank you for this suggestion. We have clarified the observation period and source of this data in the table caption (all data is derived from the 2017-2022 Planet timeseries). We have also included lake areas represented as a percentage of the total debris-covered area for each of the three area columns. We agree that this makes the data more easily comparable.**

Figure 7: I would shift this figure back to the main manuscript to highlight these results as this is a large and important point of your work and also your title refers to the decadal changes.
**Thank you for this suggestion. I believe that the track-changes version of the manuscript mistakenly pushed this figure (the 1988-2022 timeseries of SGL area on four glaciers) to the end of the document, making it appear like it was part of the supplemental material. Figure 6, 7, 8, and 9 are all meant to appear in the main manuscript, and I will ensure that this is more clear in the updated documents.**

All the best and good luck,
Anna